

# A systematic literature review on meta-heuristic based feature selection techniques for text classification

Sarah Abdulkarem Al-shalif[1], Norhalina Senan[1], Faisal Saeed[2], Wad Ghaban[3], Noraini Ibrahim[1], Muhammad Aamir[4] and Wareesa Sharif[5]

[1] Faculty of Computer Science and Information Technology, Universiti Tun Hussein Onn Malaysia, Parit Raja, Johor, Malaysia
[2] DAAI Research Group, Department of Computing and Data Science, School of Computing and Digital Technology, University of Birmingham, Birmingham, United Kingdom
[3] Applied College, University of Tabuk, Tabuk, Saudi Arabia
[4] School of Electronics, Computing and Mathematics,, University of Derby, Derby, United Kingdom
[5] Faculty of Computing, The Islamia University of Bahawalpur, Bahawalpur, Pakistan

Corresponding authors
Norhalina Senan,
halina@uthm.edu.my
Faisal Saeed, faisal.saeed@bcu.ac.uk

## ABSTRACT

Feature selection (FS) is a critical step in many data science-based applications, especially in text classification, as it includes selecting relevant and important features from an original feature set. This process can improve learning accuracy, streamline learning duration, and simplify outcomes. In text classification, there are often many excessive and unrelated features that impact performance of the applied classifiers, and various techniques have been suggested to tackle this problem, categorized as traditional techniques and meta-heuristic (MH) techniques. In order to discover the optimal subset of features, FS processes require a search strategy, and MH techniques use various strategies to strike a balance between exploration and exploitation. The goal of this research article is to systematically analyze the MH techniques used for FS between 2015 and 2022, focusing on 108 primary studies from three different databases such as Scopus, Science Direct, and Google Scholar to identify the techniques used, as well as their strengths and weaknesses. The findings indicate that MH techniques are efficient and outperform traditional techniques, with the potential for further exploration of MH techniques such as Ringed Seal Search (RSS) to improve FS in several applications.

# INTRODUCTION

The amount of high-dimensional data currently freely accessible online, such as text data, microarrays, and medical information, has drastically expanded in recent years. As a result, text classification has emerged as a crucial task in various domains, including natural language processing, information retrieval, sentiment analysis, and more. The effectiveness of text classification models heavily relies on the features used to represent the textual data. As the dimensionality of textual data increases, the need for feature selection (FS)

techniques becomes paramount to enhance model performance, reduce computational costs, and mitigate the curse of dimensionality (*Larabi Marie-Sainte & Alalyani, 2020*).

FS stands for the procedure of choosing the more important and relevant features from the datasets. Because it can increase learning accuracy, shorten learning times, and simplify learning outcomes, FS is significant and has shown significant growth. The FS approach aims to determine the most relevant and essential features to improve the classification accuracy and minimize these features without significantly affecting the performance of the classification. It increases classification effectiveness by reducing data dimensionality by removing unnecessary and redundant features (*Larabi Marie-Sainte & Alalyani, 2020*; *Mojaveriyan, Ebrahimpour-komleh & Jalaleddin, 2016*). FS technique is classified into three models: filter, wrapper, and embedded models. Filter models consider the statistical properties of the data to choose the optimal feature subset (*Bertolazzi et al., 2016*; *García-Torres et al., 2016*; *Mohanty & Das, 2018*; *Chen, Zhou & Yuan, 2019*; *Ghosh et al., 2020*). Filter models are also known as traditional FS techniques such as Information Gain, CHISquare, and ReliefF (*Tubishat et al., 2019*), they can be divided into two additional categories, namely feature ranking algorithms also known as univariate feature filters. In feature ranking algorithms, each feature is given a weight according to how relevant it is to the target concept. In contrast, subset search algorithms are known as multivariate filters (*Ghimatgar et al., 2018*). The filter model has no direct interaction with the classifier, while the wrapper model adopts optimization algorithms such as meta-heuristic (MH) that are capable of direct interaction with the features and classifier (*Tubishat et al., 2019*). In addition, embedded models interact with the classifier with a lower computational cost than the wrapper model (*Ghimatgar et al., 2018*). Due to FS's importance, many researchers have investigated its problems and proposed many techniques to improve it and remove irrelevant, redundant, and noisy features to choose a set of features that will provide the optimum accuracy and computational performance (*Kashef & Nezamabadi-pour, 2015*). Despite the growing interest in text classification and FS, there is a noticeable lack of comprehensive evaluations of meta-heuristic-based techniques specifically tailored for text classification. This review aims to bridge this gap by aggregating and critically analyzing the existing body of literature on this topic.

MH is a high-level, problem-independent algorithm framework that offers several methods for creating heuristic algorithms (*Yong, Dun-wei & Wan-qiu, 2016*). With an enormous number of features, it is computationally impossible to evaluate every state, necessitating MH search techniques. Recently, MH algorithms, such as genetic algorithms, particle swarm optimization, simulated annealing, and ant colony optimization, have shown promise in handling complex optimization challenges. These algorithms have the potential to uncover relevant features from high-dimensional text data, contributing to improved classification accuracy and model interpretability. In FS techniques, MH is used to improve the result of classification performance (*Kashef & Nezamabadi-pour, 2015*). Many researchers (*Larabi Marie-Sainte & Alalyani, 2020*; *Tubishat et al., 2019*; *Jain et al., 2019*; *Ahmad, Bakar & Yaakub, 2019*; *Al-Rawashdeh, Mamat & Hafhizah Binti Abd Rahim, 2019*; *Chantar et al., 2020*; *Kumar & Jaiswal, 2019*; *Singh & Kaur, 2020*;

*Hassonah et al., 2020*; *Gokalp, Tasci & Ugur, 2020*) attempt to utilize the advantages of the natural inspired MH search to discover the optimal subset feature to enhance classifier performance and decrease computational time and cost. These techniques make an effort to provide better solutions by using information from earlier iterations (*Kashef & Nezamabadi-pour, 2015*). The two stages of MH are exploration and exploitation, where various operators are placed to find the best solution. The potential solutions move with the search space during exploration. At the same time, the most popular regions in the search space are investigated in the exploitation. The interaction between exploration and exploitation should be balanced in a good MH, according to expectations (*Ibrahim et al., 2019*). In this review article, Meta-Heuristic Feature Selection (MH-FS) techniques have been analyzed in detail and RSS is investigated to be used as a feature subset selection technique for future direction. RSS is one of the MH techniques proposed by *Saadi et al. (2016)*. The natural behavior of the seal pup in selecting the ideal hiding lair to avoid predators served as the inspiration for RSS. As opposed to GA and PSO, when compared to its homologs, RSS is faster in locating the global optimum and maintaining the proper ratio of exploitation to exploration (*Saadi et al., 2016*). Previously, RSS is not used as FS but it can optimize the support vector machine (SVM) parameter leading to higher classification accuracy when compared to traditional SVM (*Sharif et al., 2019*).

Literature review articles might be classified into two groups: traditional literature review (TLR) and systematic literature review (SLR). TLR attends to show the research topic from a general point of view and looks at the research in general and from all directions. In comparison, SLR attempts to show the topic of the research from a broad perspective and tries to answer specific research questions through a systemic methodology (*Qasem et al., 2019*). Currently, no SLR focuses on MH-FS techniques for text classification, Therefore, this study seeks to identify the best practices, trends, and patterns in the application of MH-based feature selection techniques for text classification. By extracting insights from a diverse range of studies, researchers and practitioners can gain valuable guidance for selecting appropriate algorithms and parameters for their specific applications. As well as, this review not only evaluates the performance of various MH algorithms but also assesses their suitability for different text classification tasks, dataset characteristics, and evaluation metrics. By doing so, it contributes to the methodological advancement of text classification research. Specifically, this article focused on the SLR of FS using MH techniques published in the period from 2015 to 2022. This SLR aims to summarize and clarify available guidance related to (1) the MH techniques for FS techniques, (2) the MH-FS techniques that can be used for text classification, (3) the comparisons of the performance of the MH techniques over the traditional techniques, and (4) highlight the strengths and weaknesses of different MH techniques. While, the intended audience for this SLR encompasses a wide range of individuals with a shared interest in advancing the capabilities of text classification through the integration of MH-based feature selection techniques such as academics, data scientists, students, industry practitioners, decision-makers, and researchers in FS, text classification, and MH fields. Whether seeking theoretical insights, practical guidance, or interdisciplinary connections,

this review offers a valuable resource to foster informed decision-making, research, and innovation in the dynamic field of text analysis.

The rest of the article is organized as follows. "Survey Methodology" explains the methodology used in this review. "Bibliometric Analysis by Co-occurrence (Authors Keywords)" presents the bibliometric analysis maps. "Results and Discussion" presents and discusses the review results. The conclusion and future work are presented in "Conclusion".

## SURVEY METHODOLOGY

In this article, the planning, conducting, and reporting procedures are according to the procedure given by Kitchenham and Charters (*El-Gohary, Nasr & Wahaab, 2000*). The procedures consist of five steps which are: identifying research questions, search strategy and study selection criteria, quality assessment criteria, data extraction, and data synthesis processes (*Qasem et al., 2019*).

Firstly, the research questions were meticulously crafted to address the inherent challenges associated with FS-MH techniques. Subsequently, the search strategy has been elucidated, encompassing the identification of search terms, utilization of search resources, and the systematic execution of the search process aimed at identifying and selecting pertinent studies. The selection of relevant studies, in alignment with the research questions, was carried out based on well-defined inclusion and exclusion criteria. The next phase involved articulating quality assessment criteria, a pivotal component for rigorously analyzing and assessing the studies under consideration. Finally, the processes of data extraction and synthesis was detailed, which form the concluding steps in this SLR. The ensuing subsections provide an in-depth exposition of each of these procedural steps, and Fig. 1 offers a visual representation of the sequential flow of the SLR methodology.

### Research questions

This SLR is purposed to define the guidance gained from the previous studies using the MH techniques for FS in text classification. Table 1 presents five research questions discussed in this SLR. From the previous studies, MH techniques that have been used for FS were identified in (RQ1). The analysis of these studies has been conducted to answer the questions. RQ2 determines which FS-MH techniques have been used for text classification. RQ3 compares the performance of MH-FS techniques with the traditional techniques. The purpose of this question is to discover if MH methods are superior to conventional methods. RQ4 identified the advantages and limitations of different MH techniques to guide the selection of appropriate MH techniques, while the final question (RQ5) investigated how RSS can be used as FS.

### Search strategy and study selection

Detailed analysis of this step is described in four subsections which are search terms, literature sources, search process, and study selection.

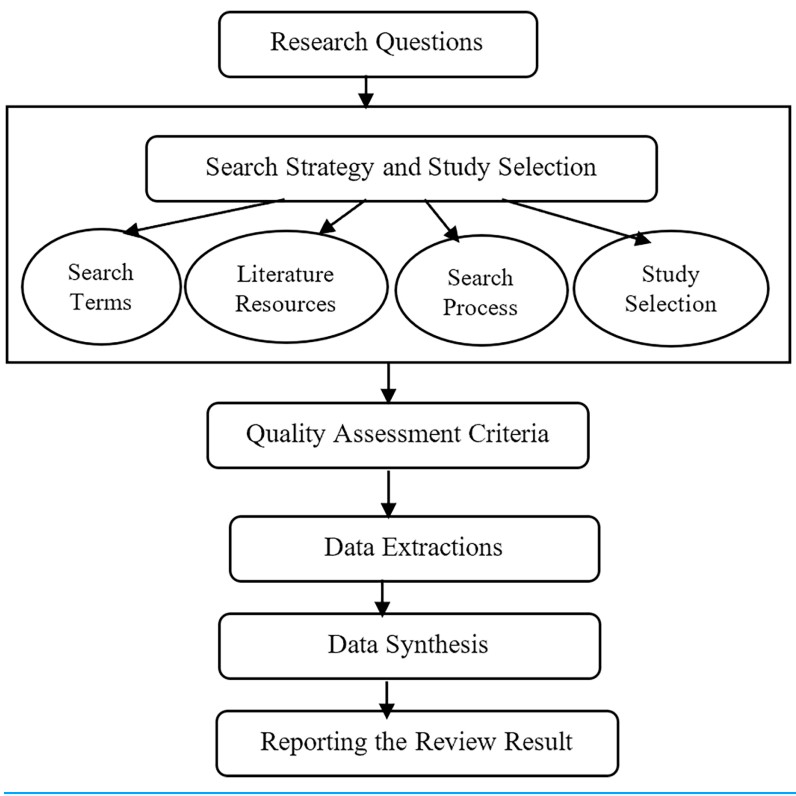

**Figure 1** **The steps of the SLR.**

| RQ# | Research questions | Reasons |
|---|---|---|
| **Table 1** **Research questions.** | | |
| RQ1 | Which metaheuristic (MH) techniques have been utilized for feature selection (FS) | Identify which MH techniques are commonly used for FS. |
| RQ2 | In the context of text classification, which specific metaheuristic techniques have been applied for feature selection? | Determine which MH techniques are commonly used for FS in text classification. |
| RQ2.1 | What are the datasets employed in the application of metaheuristic feature selection (MH-FS) for text classification? | Identify common datasets used for text classification. |
| RQ2.2 | Which classifiers have been used with MH-FS in text classification? | Identify common classifiers used for text classification. |
| RQ2.3 | Which performance evaluation metrics are commonly utilized to assess the effectiveness of MH-FS in text classification? | Identify evaluation metrics reported to be appropriate for text classification. |
| RQ3 | Is there empirical evidence indicating that MH-FS techniques outperform traditional FS methods in the domain of text classification? | Investigate the performance of the MH techniques over the traditional techniques |
| RQ4 | What are the discernible strengths and weaknesses of MH techniques in the context of FS? | Highlighted the strengths and weaknesses of different MH techniques |
| RQ5 | How can the RSS be effectively leveraged as FS technique? | Investigate the application of the RSS-FS technique algorithm. |

## Search terms

Five steps were conducted to extract the search terms which are as follows (*Malhotra, 2015*):

1) Extract the key terms from the research questions.

2) Determine the synonyms and alternative spellings for the main terms.

3) Explore the keywords and terminology from existing research articles.

4) Combine the synonyms and alternative spellings using the Boolean operator "OR".

5) Connect the main terms using the Boolean operator "AND".

All research terms were derived from the explored topic. These terms are feature selection, attribute selection, text classification, text categorization, meta-heuristics, and metaheuristics. The final search terms that connect with the Boolean operators were as follows: (("feature selection" OR "attribute selection") AND ("text classification" OR "text categorization") AND ("meta-heuristics" OR "metaheuristics")).

## Literature sources

The relevant studies were investigated through the Scopus, Science Direct, and Google Scholar databases. The records identified in the period from 2008 to 2022 used "feature selection" and "attribute selection" as the main keywords and the rest of the keywords to specify and limit the selected studies.

## Search process

The number of studies obtained from the Scopus, Science Direct, and Google Scholar databases using the main keywords were as follows: 45,236 for Scopus, 53,030 for Science Direct, and 19,300 for Google Scholar. These studies were then further filtered using specific keywords such as "text classification" or "text categorization," resulting in 5,186 studies for Scopus, 1,922 studies for Science Direct, and 842 studies for Google Scholar. Another set of specific keywords, namely "meta-heuristic" or "metaheuristic," yielded 226 studies for Scopus, 438 studies for Science Direct, and 34 studies for Google Scholar. After the initial search, the relevant studies were chosen according to predefined inclusion and exclusion criteria explained in the following subsection.

## Study selection

The studies obtained were further narrowed down to include only articles published between 2015 and 2022. The resulting numbers were 194 studies for Scopus, 200 studies for Science Direct, and 25 studies for Google Scholar. The Inclusion-Exclusion criterion was then applied to limit the search scope. The purpose was to evaluate all selected studies that either facilitated or directly addressed at least one research question in the field. Additionally, the analysis focused on articles and review articles, and only English articles were included. Based on these criteria, the list of studies was reduced to 143 for Scopus, 38 for Science Direct, and 18 for Google Scholar. This initial list was further analyzed and filtered by examining the titles, keywords, and abstracts to remove irrelevant papers. The systematic literature review (SLR) was conducted on research published between 2015 and

**Table 2 Inclusion and exclusion criteria.**

| Inclusion criteria | Exclusion criteria |
|---|---|
| English papers | Any other language papers |
| All paper discussing MH in FS. | Paper that did not have any link with the research question |
| Papers that can answer at least one research question | Papers with the same author and the same MH techniques because this information tended to be duplicated |
| Papers (≥ 3 pages) | Short papers (<3 pages) |
| Article, and review papers | Conference, books, and chapter in book |

2022. The final list consisted of 91 studies for Scopus, 15 for Science Direct, and five for Google Scholar. Furthermore, any duplicate papers from Science Direct and Google Scholar were removed, resulting in a final list of 91, 12, and five studies for all databases. In addition, there are 15 selected studies were chosen out of limitation. Table 2 presents the details of the inclusion and exclusion criteria.

## Quality assessment criteria

The selected studies were assessed with great care and seriousness to maintain a high-quality standard. This assessment included evaluating the novelty of the proposed techniques and the ability of the studies to address at least one research question. Furthermore, special consideration was given to choosing high-quality studies from high-impact journals available in the Scopus, Science Direct, and Google Scholar digital libraries. These measures were taken to ensure a thorough quality check and maintain the overall standard of the selected studies.

## Data extraction

During the data extraction phase, the selected studies were utilized to gather crucial information necessary to tackle the research questions. The extracted information encompassed various aspects such as author details, publication year, applied meta-heuristic (MH) techniques, utilized datasets, employed classifiers, performance measures used for evaluation, and the obtained results. This collected information was then organized and analyzed to facilitate further data synthesis.

## Data synthesis

The data synthesis phase in the SLR involves summarizing and interpreting the collected information from the selected studies. This phase aims to address the research questions through analysis, discussion, and various forms of representation such as tables, graphs, charts, *etc*. The SLR processes are typically executed multiple times to ensure an effective review process that yields the most relevant and suitable studies. The process of this particular SLR began with the identification of research questions, followed by the application of relevant search terms on the Scopus, Science Direct, and Google Scholar digital libraries. This initial search yielded 143, 38, and 18 research papers, respectively. After analyzing and filtering the studies based on inclusion and exclusion criteria, as well as

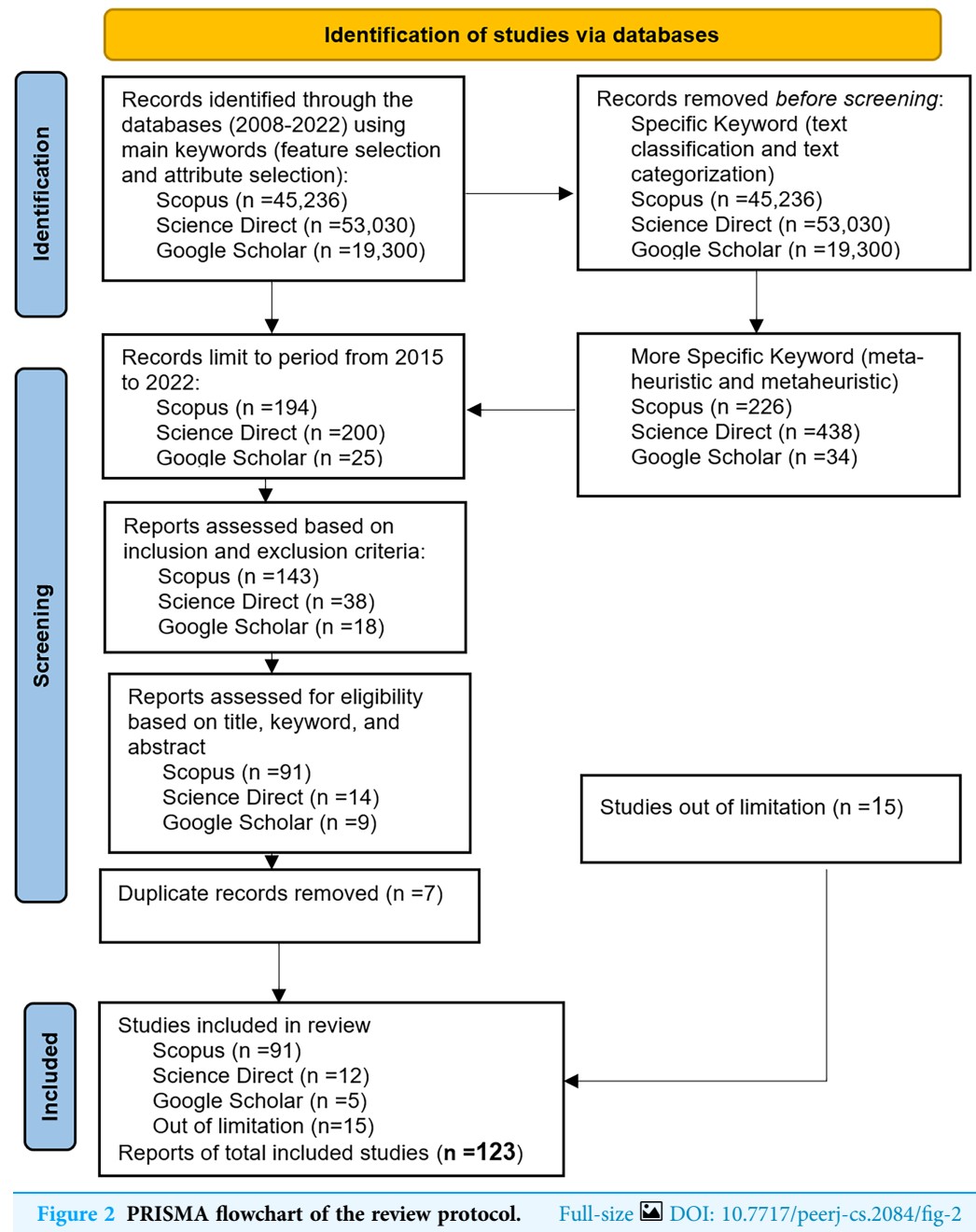

**Figure 2 PRISMA flowchart of the review protocol.**

removing any duplicated papers, the final number of obtained studies was 91, 12, and five, a total of 108 articles. Figure 2 illustrates the search steps protocol using the PRISMA flowchart, highlighting the progression from the initial search to the final selection of studies. After thoroughly searching through the selected studies, a total of 123 studies were found to be valuable for this SLR, as they exhibited a high level of relevance in addressing the chosen research domain.

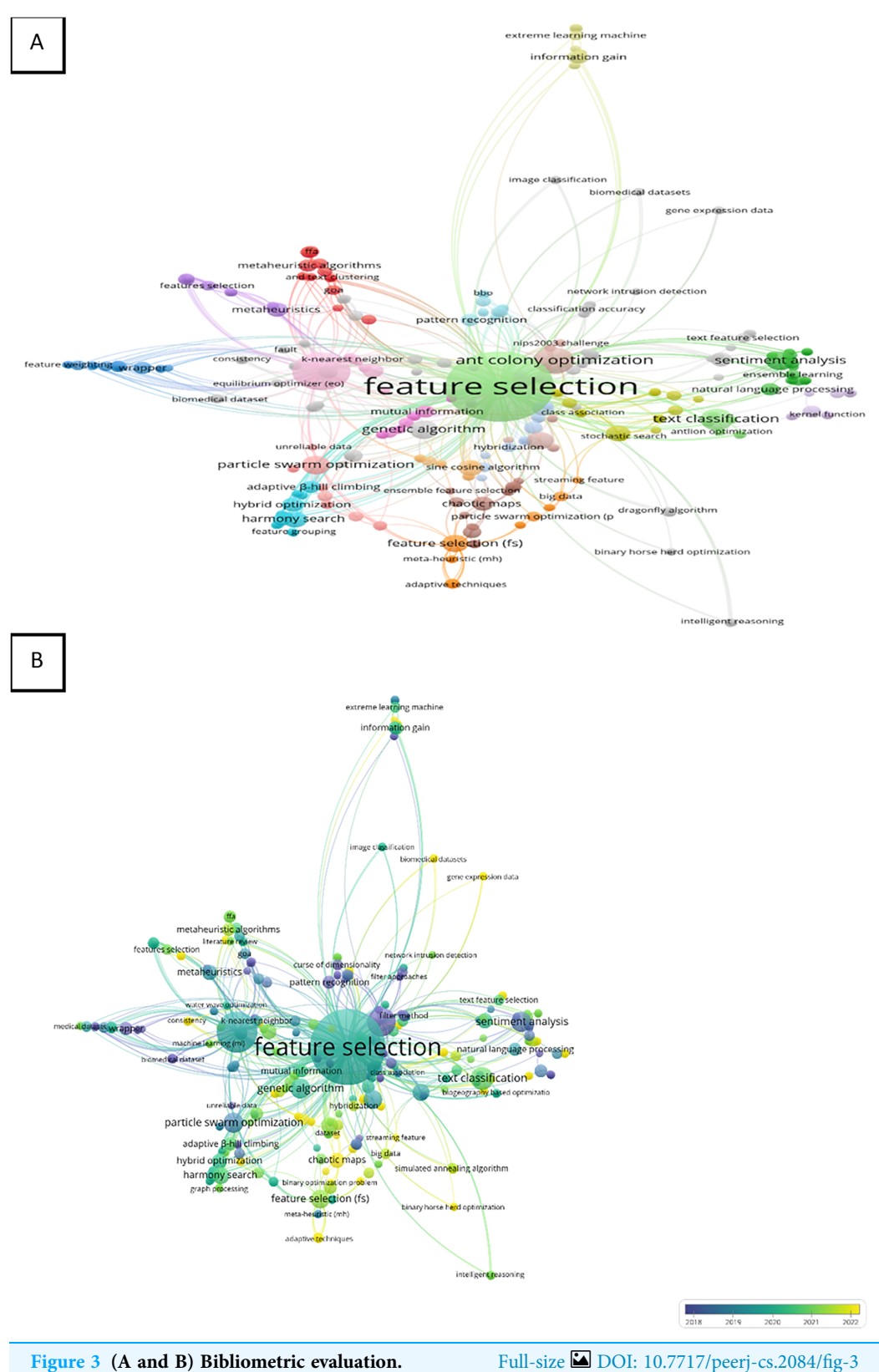

**Figure 3** (A and B) Bibliometric evaluation.

# BIBLIOMETRIC ANALYSIS BY CO-OCCURRENCE (AUTHORS KEYWORDS)

The keywords used by the authors of the article which occurred one time or more in the Scopus database were joined in the final analysis. The number of keywords used by the authors in the analysis was 417. The top three keywords that appeared were "feature selection", "classification" and "ant colony optimization" with the following total number of occurrences 96, 28, and 14 including total strength 419, 152, and 54, respectively. Therefore, the network visualization demonstrates the three top keywords with higher weights of items, larger labels, and circles as presented in Fig. 3B. The keywords were presented in the map into 37 clusters with 788 of total length strength as shown in Fig. 3A. This finding can represent the wide applications of these keywords. On the other hand, the total occurrence of "text classification" and "metaheuristic" were three and two with total length strengths of 10 and 8, respectively.

Overly visualization map Fig. 3B determined the colors of the items, where blue (lowest score), green (middle score), and yellow (highest score) are the standard colors. The map shows that "feature selection" and "classification" were mostly used in the period between 2019 to 2021 however "ant colony optimization" was highly used between 2018 and 2019. In addition, "metaheuristic" was applied in 2021 more than any year before, and "text classification" was recently applied especially in 2020 as present in the map's color and networks Fig. 3B.

# RESULTS AND DISCUSSION

The primary contribution of this SLR is the systematic analysis of MH techniques in FS from 2015 to 2022, based on 108 primary studies from databases such as Scopus, Science Direct, and Google Scholar. The review highlights the efficiency of MH techniques compared to traditional ones and suggests the potential for further exploration of techniques like the Ringed Seal Search (RSS) to enhance FS in various applications. This section focuses on the results and findings of the review. Firstly, it provides a brief overview of the chosen studies. Subsequently, a separate subsection delves into a detailed discussion of the findings that address the research questions.

## Overview of the selected study

After conducting a scan and filtering process on studies published between 2015 and 2022, a total of 108 relevant studies were initially obtained. However, 15 studies were out of limitations and were included. Therefore, the final number of studies included in this review amounted to 123. Figure 4 visually represents the distribution of these studies over the period from 2015 to 2022, indicating that the highest number of studies was observed in 2019, while the lowest number of studies occurred in 2017.

The quality of the selected studies was assessed by considering the quality and impact factors of the journals in which they were published. This ensured that high-quality studies were included. Table 3 presents a comprehensive list of the journals that published the selected studies, along with the corresponding number of studies in each journal. It also provides information on the Quartile ranking of these journals in the International

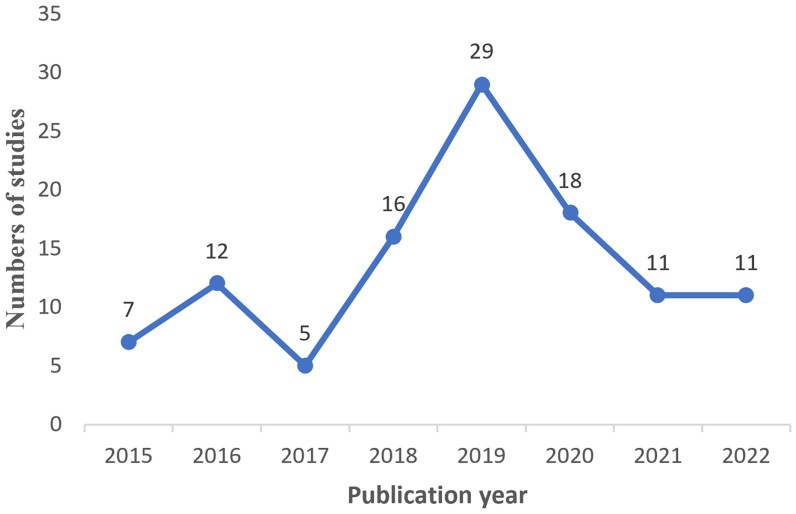

Figure 4 Studies distribution by year of publication.

Table 3 Summary of the journals of selected studies.

| | Publication journals | # of studies | Q# in ISI (2021) | Impact factor in ISI (2021) | Q# in Scopus indexed (2021) | SJR in Scopus (2021) |
|---|---|---|---|---|---|---|
| 1 | Artificial intelligence review | 1 | Q1 | 9.588 | Q1 | 2.49 |
| 2 | Expert systems with applications | 9 | Q1 | 8.665 | Q1 | 1.87 |
| 3 | Pattern recognition | 3 | Q1 | 8.518 | Q1 | 2.09 |
| 4 | Applied soft computing | 8 | Q1 | 8.263 | Q1 | 1.88 |
| 5 | Information sciences | 2 | Q1 | 8.233 | Q1 | 2.29 |
| 6 | Knowledge-based systems | 10 | Q1 | 8.139 | Q1 | 2.07 |
| 7 | Engineering applications of artificial intelligence | 2 | Q1 | 7.802 | Q1 | 1.73 |
| 8 | European journal of operational research | 2 | Q1 | 6.363 | Q1 | 2.37 |
| 9 | Computer networks | 1 | Q1 | 5.493 | Q1 | 1.36 |
| 10 | Neurocomputing | 5 | Q2 | 5.779 | Q1 | 1.48 |
| 11 | Neural computing and applications | 6 | Q2 | 5.102 | Q1 | 1.17 |
| 12 | Applied intelligence | 5 | Q2 | 5.019 | Q2 | 1.15 |
| 13 | International journal of approximate reasoning | 1 | Q2 | 4.452 | Q1 | 0.98 |
| 14 | International journal of machine learning and cybernetics | 1 | Q2 | 4.377 | Q1 | 1 |
| 15 | ACM transactions on knowledge discovery from data | 1 | Q2 | 4.157 | Q1 | 1.27 |
| 16 | PLoS ONE | 2 | Q2 | 3.752 | Q1 | 0.89 |
| 17 | Soft computing | 4 | Q2 | 3.732 | Q2 | 0.82 |
| 18 | Journal of ambient intelligence and humanized computing | 2 | Q2 | 3.662 | Q1 | 0.91 |
| 19 | IEEE access | 8 | Q2 | 3.476 | Q1 | 0.93 |
| 20 | Arabian journal for science and engineering | 1 | Q2 | 2.807 | Q2 | 0.48 |
| 21 | Connection science | 1 | Q2 | 0.641 | Q2 | 0.85 |

(Continued)

| | Publication journals | # of studies | Q# in ISI (2021) | Impact factor in ISI (2021) | Q# in Scopus indexed (2021) | SJR in Scopus (2021) |
|---|---|---|---|---|---|---|
| 22 | Medical & biological engineering & computing | 1 | Q3 | 3.079 | Q2 | 0.65 |
| 23 | Applied artificial intelligence | 2 | Q3 | 2.777 | Q3 | 0.49 |
| 24 | Multimedia tools and applications | 1 | Q3 | 2.577 | Q1 | 0.72 |
| 25 | Knowledge and information systems | 2 | Q3 | 2.531 | Q2 | 0.77 |
| 26 | Cluster computing | 2 | Q3 | 2.303 | Q2 | 0.62 |
| 27 | Geotechnical and geological engineering | 1 | Q3 | 0.45 | Q1 | 0.51 |
| 28 | Pertanika journal of science and technology | 1 | Q3 | 0.13 | Q3 | 0.19 |
| 29 | Intelligent data analysis | 2 | Q4 | 1.321 | Q3 | 0.38 |
| 30 | International arab journal of information technology | 1 | Q4 | 0.967 | Q3 | 0.31 |
| 31 | IEICE TRANSACTIONS on information and systems | 1 | Q4 | 0.695 | Q3 | 0.28 |
| 32 | Journal of medical imaging and health informatics | 1 | Q4 | 0.659 | Q4 | 0.19 |
| 33 | Journal of king saud university—computer and information sciences | 6 | – | – | Q1 | 0.92 |
| 34 | International journal of network security | 1 | – | – | Q2 | 0.336 |
| 35 | Informatics in medicine unlocked | 1 | – | – | Q2 | 0.79 |
| 36 | IISE transactions on healthcare systems engineering | 1 | – | – | Q2 | 0.4 |
| 37 | International journal of intelligent systems and applications | 1 | – | – | Q3 | 0.241 |
| 38 | Electronic notes in discrete mathematics | 1 | – | – | Q4 | 0.11 |
| 39 | International journal of circuits, systems and signal processing | 1 | – | – | Q4 | 0.156 |
| 40 | Journal of telecommunication, electronic and computer engineering | 1 | – | – | Q4 | 0.152 |
| 41 | Future internet | 1 | – | – | Q2 | 0.77 |

Scientific Indexing (ISI) and their impact factors. Additionally, the Quartile ranking in Scopus is indexed with their Scientific Journal Ranking (SJR). It can be concluded that the ISI journals accounted for 85.19% of the total number of selected studies, further confirming the overall quality of the included literature.

The selected studies were assigned unique identifiers (IDs) for easy reference and consistency throughout the subsequent subsections. Table 4 presents a comprehensive list of all the studies included in the review, along with their corresponding IDs and references. Furthermore, Table 5 presents a summary of the selected studies' IDs along with the research questions they have addressed. The analysis reveals that all the selected studies have contributed to answering RQ1. For RQ2, 28 studies have provided relevant insights. Similarly, 26 studies have addressed RQ3, while 40 studies have tackled RQ4. It is worth mentioning that one paper, which was outside the predefined limitations, addressed RQ5 by utilizing RSS to enhance SVM in text classification. More detailed information on these findings can be found in the subsequent subsections.

**Table 4 Selected primary studies along with their IDs and references.**

| ID | Paper author | Ref. | ID | Paper author | Ref. | ID | Paper author | Ref. |
|---|---|---|---|---|---|---|---|---|
| RP1 | Tabakhi (2015a) | *Tabakhi & Moradi (2015)* | RP2 | Wang (2015) | *Wang et al. (2015)* | RP3 | Tabakhi (2015b) | *Tabakhi et al. (2015)* |
| RP4 | Moshki (2015) | *Moshki, Kabiri & Mohebalhojeh (2015)* | RP5 | Moradi (2015) | *Moradi & Rostami (2015)* | RP6 | Inbarani (2015) | *Inbarani, Bagyamathi & Azar (2015)* |
| RP7 | Kashef (2015) | *Kashef & Nezamabadi-pour (2015)* | RP8 | Zorarpacı (2016) | *ZorarpacI & Özel (2016)* | RP9 | Zawbaa (2016) | *Zawbaa, Emary & Grosan (2016)* |
| RP10 | Zarshenas (2016) | *Zarshenas & Suzuki (2016)* | RP11 | Bertolazzi (2016) | *Bertolazzi et al. (2016)* | RP12 | Dadaneh (2016) | *Dadaneh, Markid & Zakerolhosseini (2016)* |
| RP13 | Das (2016) | *Das, Mishra & Shaw (2016)* | RP14 | Yong (2016) | *Yong, Dun-wei & Wan-qiu (2016)* | RP15 | Saraswathi (2016) | *Saraswathi & Tamilarasi (2016)* |
| RP16 | Salama (2016) | *Salama, Abdelbar & Anwar (2016)* | RP17 | Mojaveriyan (2016) | *Mojaveriyan, Ebrahimpour-komleh & Jalaleddin (2016)* | RP18 | Emary (2016) | *Emary & Zawbaa (2016)* |
| RP19 | Garcia-Torres (2016) | *García-Torres et al. (2016)* | RP20 | Ahmad (2017) | *Ahmad et al. (2017)* | RP21 | Barani (2017) | *Barani, Mirhosseini & Nezamabadi-pour (2017)* |
| RP22 | Deniz (2017) | *Deniz et al. (2017)* | RP23 | Zhang (2017) | *Zhang, Song & Gong (2017)* | RP24 | Kuo (2018) | *Kuo et al. (2018)* |
| RP25 | Costa (2018) | *Costa et al. (2018)* | RP26 | Ghimatgar (2018) | *Ghimatgar et al. (2018)* | RP27 | Jadhav (2018) | *Jadhav, He & Jenkins (2018)* |
| RP28 | Javidi (2018) | *Javidi & Zarisfi Kermani (2018)* | RP29 | Kiziloz (2018) | *Kiziloz et al. (2018)* | RP30 | Mafarja (2018a) | *Mafarja & Mirjalili (2018)* |
| RP31 | Mafarja (2018b) | *Mafarja et al. (2018)* | RP32 | Mohanty (2018) | *Mohanty & Das (2018)* | RP33 | Oztekin (2018) | *Oztekin et al. (2018)* |
| RP34 | Rais (2018) | *Rais & Mehmood (2018)* | RP35 | Sayed (2018) | *Sayed, Khoriba & Haggag (2018)* | RP36 | Singh (2018) | *Singh & Singh (2018)* |
| RP37 | Yelmen (2018) | *Yelmen et al. (2018)* | RP38 | Cheruku (2018) | *Cheruku et al. (2018)* | RP39 | Abd El Aziz (2018) | *Aziz & Hassanien (2018)* |
| RP40 | Ahmadi (2019) | *Ahmadi et al. (2019)* | RP41 | Jain (2019) | *Jain et al. (2019)* | RP42 | Ahmad (2019) | *Ahmad, Bakar & Yaakub (2019)* |
| RP43 | Al-Rawashdeh (2019) | *Al-Rawashdeh, Mamat & Hafhizah Binti Abd Rahim (2019)* | RP44 | Sayed (2019) | *Sayed, Hassanien & Azar (2019)* | RP45 | Thiyagarajan (2019) | *Thiyagarajan & Shanthi (2019)* |
| RP46 | Ghosh (2019) | *Ghosh et al. (2019)* | RP47 | Arora (2019a) | *Arora et al. (2019)* | RP48 | Arora (2019b) | *Arora & Anand (2019)* |
| RP49 | Chantar (2019) | *Chantar et al. (2020)* | RP50 | Chen (2019) | *Chen, Zhou & Yuan (2019)* | RP51 | Dash (2019) | *Dash, Dash & Rautray (2019)* |
| RP52 | Ghosh (2019) | *Ghosh et al. (2020)* | RP53 | Han (2019) | *Han, Zhou & Zhou (2019)* | RP54 | Mafarja (2019a) | *Mafarja & Mirjalili (2019)* |
| RP55 | Tubishat (2019) | *Tubishat et al. (2019)* | RP56 | Hichem (2019) | *Hichem et al. (2019)* | RP57 | Ibrahim (2019) | *Ibrahim et al. (2019)* |
| RP58 | Kumar (2019) | *Kumar & Jaiswal (2019)* | RP59 | Liang (2019) | *Liang, Wang & Liu (2019)* | RP60 | Mafarja (2019b) | *Mafarja et al. (2019)* |

(Continued)

| ID | Paper author | Ref. | ID | Paper author | Ref. | ID | Paper author | Ref. |
|---|---|---|---|---|---|---|---|---|
| RP61 | Manbari (2019) | *Manbari, AkhlaghianTab & Salavati (2019)* | RP62 | Krishnan (2019) | *Krishnan & Sowmya Kamath (2019)* | RP63 | Selvarajan (2019) | *Selvarajan, Jabar & Ahmed (2019)* |
| RP64 | Singh (2020) | *Singh & Kaur (2020)* | RP65 | Xue (2019) | *Xue, Xue & Zhang (2019)* | RP66 | Zakeri (2019) | *Zakeri & Hokmabadi (2019)* |
| RP67 | Malar (2019) | *Malar, Nadarajan & Gowri Thangam (2019)* | RP68 | Zhu (2019) | *Zhu et al. (2019)* | RP69 | Hassonah (2020) | *Hassonah et al. (2020)* |
| RP70 | Hu (2020) | *Hu, Pan & Chu (2020)* | RP71 | Bhattacharyya (2020) | *Bhattacharyya et al. (2020)* | RP72 | Oliva (2020) | *Oliva & Elaziz (2020)* |
| RP73 | Too (2020) | *Too & Rahim Abdullah (2020)* | RP74 | Arora (2020) | *Arora, Sharma & Anand (2020)* | RP75 | Anand (2020) | *Anand & Arora (2020)* |
| RP76 | Tawhid (2020) | *Tawhid & Ibrahim (2020)* | RP77 | Anter (2020) | *Anter & Ali (2020)* | RP78 | Marie-Sainte (2020) | *Larabi Marie-Sainte & Alalyani (2020)* |
| RP79 | Gokalp (2020) | *Gokalp, Tasci & Ugur (2020)* | RP80 | Ibrahim (2020) | *Ibrahim, Tawhid & Ward (2020)* | RP81 | Tubishat (2020) | *Tubishat et al. (2020)* |
| RP82 | Pan (2021) | *Pan et al. (2021)* | RP83 | Mohan (2021) | *Mohan & Moorthi (2021)* | RP84 | Sharaff (2021) | *Sharaff et al. (2021)* |
| RP85 | Abualigah (2021) | *Abualigah & Dulaimi (2021)* | RP86 | Ma (2021) | *Ma et al. (2021)* | RP87 | Tubishat (2022) | *Tubishat et al. (2022)* |
| RP88 | Osmani (2022) | *Osmani, Mohasefi & Gharehchopogh (2022)* | RP89 | Das (2022) | *Das, Naik & Behera (2022)* | RP90 | Feng (2022) | *Feng, Kuang & Zhang (2022)* |
| RP91 | Zhao (2022) | *Zhao et al. (2022)* | RP92 | Hosseinalipour (2022) | *Hosseinalipour & Ghanbarzadeh (2022)* | RP93 | Pashaei (2017) | *Pashaei & Aydin (2017)* |
| RP94 | Hammouri (2020) | *Hammouri et al. (2020)* | RP95 | Souza (2020) | *Thom de Souza et al. (2020)* | RP96 | Purushothaman (2020) | *Purushothaman, Rajagopalan & Dhandapani (2020)* |
| RP97 | Agrawal (2020) | *Agrawal, Kaur & Sharma (2020)* | RP98 | Sadeghian (2021) | *Sadeghian, Akbari & Nematzadeh (2021)* | RP99 | Dash (2021) | *Dash (2021)* |
| RP100 | Paul (2021) | *Paul et al. (2021)* | RP101 | Wang (2022) | *Wang et al. (2022)* | RP102 | Eluri (2022) | *Eluri & Devarakonda (2022)* |
| RP103 | Allam (2022) | *Allam & Nandhini (2022)* | RP104 | Liu (2022) | *Liu et al. (2022)* | RP105 | Pandey (2020) | *Pandey, Rajpoot & Saraswat (2020)* |
| RP106 | Ansari (2021) | *Ansari et al. (2021)* | RP107 | Albashish (2021) | *Albashish et al. (2021)* | RP108 | Al-Dyani (2022) | *Al-Dyani, Ahmad & Kamaruddin (2022)* |

**Table 5 RQs addressed in individual study.**

| Paper ID | RQ1 | RQ2 | RQ3 | RQ4 | Paper ID | RQ1 | RQ2 | RQ3 | RQ4 | Paper ID | RQ1 | RQ2 | RQ3 | RQ4 |
|---|---|---|---|---|---|---|---|---|---|---|---|---|---|---|
| RP1 | √ | – | √ | √ | RP38 | √ | – | – | – | RP75 | √ | – | – | √ |
| RP2 | √ | √ | √ | √ | RP39 | √ | – | – | √ | RP76 | √ | – | – | – |
| RP3 | √ | – | √ | √ | RP40 | √ | – | – | √ | RP77 | √ | – | – | √ |
| RP4 | √ | – | – | √ | RP41 | √ | √ | √ | – | RP78 | √ | √ | √ | – |
| RP5 | √ | – | √ | √ | RP42 | √ | √ | √ | √ | RP79 | √ | √ | √ | – |
| RP6 | √ | – | √ | √ | RP43 | √ | √ | – | √ | RP80 | √ | – | – | – |
| RP7 | √ | – | – | √ | RP44 | √ | – | – | √ | RP81 | √ | – | – | √ |
| RP8 | √ | – | √ | √ | RP45 | √ | √ | – | √ | RP82 | √ | – | – | – |
| RP9 | √ | – | – | √ | RP46 | √ | – | √ | – | RP83 | √ | √ | – | – |
| RP10 | √ | – | √ | – | RP47 | √ | – | – | √ | RP84 | √ | √ | – | – |
| RP11 | √ | – | – | – | RP48 | √ | – | – | √ | RP85 | √ | – | – | – |
| RP12 | √ | – | √ | √ | RP49 | √ | √ | – | √ | RP86 | √ | – | – | – |
| RP13 | √ | – | √ | √ | RP50 | √ | – | √ | √ | RP87 | √ | √ | – | – |
| RP14 | √ | – | – | – | RP51 | √ | – | – | √ | RP88 | √ | √ | – | – |
| RP15 | √ | √ | – | – | RP52 | √ | – | – | – | RP89 | √ | – | – | – |
| RP16 | √ | – | – | – | RP53 | √ | – | – | √ | RP90 | √ | – | – | – |
| RP17 | √ | √ | √ | – | RP54 | √ | – | – | √ | RP91 | √ | – | – | – |
| RP18 | √ | – | – | – | RP55 | √ | √ | – | √ | RP92 | √ | √ | – | – |
| RP19 | √ | √ | – | – | RP56 | √ | – | – | – | RP93 | √ | √ | – | √ |
| RP20 | √ | √ | – | – | RP57 | √ | – | – | √ | RP94 | √ | – | – | – |
| RP21 | √ | – | – | – | RP58 | √ | √ | √ | – | RP95 | √ | – | – | – |
| RP22 | √ | – | – | – | RP59 | √ | – | – | – | RP96 | √ | √ | – | – |
| RP23 | √ | – | √ | √ | RP60 | √ | – | √ | – | RP97 | √ | √ | – | – |
| RP24 | √ | – | – | – | RP61 | √ | – | √ | √ | RP98 | √ | – | – | – |
| RP25 | √ | – | – | – | RP62 | √ | – | √ | – | RP99 | √ | – | – | – |
| RP26 | √ | – | √ | √ | RP63 | √ | – | – | – | RP100 | √ | √ | – | – |
| RP27 | √ | – | – | – | RP64 | √ | √ | – | – | RP101 | √ | – | – | – |
| RP28 | √ | – | – | √ | RP65 | √ | – | – | – | RP102 | √ | – | – | – |
| RP29 | √ | – | – | √ | RP66 | √ | – | – | √ | RP103 | √ | – | – | – |
| RP30 | √ | – | √ | – | RP67 | √ | – | – | – | RP104 | √ | – | – | – |
| RP31 | √ | – | √ | – | RP68 | √ | – | – | √ | RP105 | √ | – | – | – |
| RP32 | √ | – | – | – | RP69 | √ | √ | √ | – | RP106 | √ | – | – | – |
| RP33 | √ | – | – | – | RP70 | √ | – | – | √ | RP107 | √ | – | – | – |
| RP34 | √ | – | √ | – | RP71 | √ | – | – | – | RP108 | √ | √ | – | – |
| RP35 | √ | – | – | – | RP72 | √ | – | – | √ | | | | | |
| RP36 | √ | √ | – | √ | RP73 | √ | – | – | – | | | | | |
| RP37 | √ | √ | – | – | RP74 | √ | – | – | – | | | | | |

### RQ1: which metaheuristic (MH) techniques have been utilized for feature selection (FS)?

In this section, the focus is on discussing and identifying the MH techniques that have been utilized for feature selection in various machine learning problems, including pattern recognition, email classification, microarray data classification, sentiment analysis, and text classification. The findings from the primary selected studies reveal that all the studies have provided insights into (RQ1). These studies have classified the MH techniques into three main groups according to their sources of inspiration. These groups are Evolutionary Algorithms (EA), Physics-Based (PB) Algorithms, and Swarm Intelligence (SI) Algorithms. This categorization provides a broad understanding of the different types of MH techniques employed for FS across various domains (*Kumar & Bawa, 2020*).

Evolutionary Algorithms (EAs) draw inspiration from the natural processes of evolution. One of the commonly used algorithms in this category is the Genetic Algorithm (GA) belongs to a class of optimization algorithms that draw inspiration from the principles of natural selection and genetics. They mimic the principles of evolution to solve complex problems by iteratively searching and refining a population of potential solutions, that has been utilized for feature selection in numerous studies. Specifically, in RP13, RP22, RP27, RP32, RP33, RP37, RP40, RP46, RP62, RP85, and RP106. Another algorithm in the EA category is Differential Evolution (DE), which is used to solve continuous optimization problems, and was utilized for feature selection in RP8 and RP55. Evolutionary Population Dynamics (EPD) combines concepts from evolutionary biology and population dynamics with computational methods, it was used in RP31, Imperialist Competitive Algorithm (ICA) draws inspiration from the socio-political behavior of imperialistic systems, it was applied in RP17 and RP88. RP95 utilized the Binary Coyote Optimization Algorithm (BCOA), which draws inspiration from the intelligent behavior exhibited by coyotes in their natural environment. The Golden Eagle Optimizer (GEO) inspired by the behavior and characteristics of golden eagles in nature, is used by RP102. Biogeography-based Optimization (BBO) is influenced by the principles of biogeography, which involves the study of the distribution of biological organisms across different geographic regions, it is used by RP107. These studies demonstrate the application of different evolutionary algorithms for feature selection in various machine-learning problems.

Secondly, is the Physics-Based (PBs) Algorithms, which aim to mimic physical rules in their search process. Several common algorithms in this category have been employed for feature selection in the selected studies. Simulated Annealing (SA) draws inspiration from the annealing process used in metallurgy, it was utilized in RP4, RP28, RP43, and RP87. The Harmony Search (HS) algorithm takes inspiration from the musical improvisation process, it was applied in RP2, RP6, RP13, RP71, and RP99. The Gravitational Search Algorithm (GSA) derives its inspiration from the fundamental law of gravity and the motion of celestial bodies, it was used in RP21, RP28, and RP68. The Teaching Learning Based Optimization (TLBO) algorithm draws inspiration from the teaching and learning processes that take place in a classroom, it was employed in RP29 and RP103. Water Cycle algorithms (WCA) were utilized in RP43. Atom Search Optimization (ASO) was applied in

RP73. The multi-Verse Optimizer (MVO) algorithm inspired by the concept of multiple universes and parallel universes in theoretical physics, was used in RP69. The Interior Search Algorithm (ISA) designed for solving constrained optimization problems, was employed in RP74. Lastly, RP93 utilizes the Black Hole Optimization (BHO) algorithm, a robust stochastic optimization technique that takes inspiration from the behavior of black holes in outer space. The key distinction between EAs and PBs lies in the mechanism of communication and movement of search agents within the search space. PB algorithms rely on physical rules to guide the search process, while EAs are inspired by evolutionary processes. This difference in approach enables PB algorithms to explore the search space using physics-inspired mechanisms.

The third group consists of Swarm intelligence (SI) algorithms, which draw inspiration from the collective behavior observed in swarms, herds, flocks, or schools of living organisms in nature. While these algorithms share similarities with evolutionary algorithms (EAs) and population-based (PB) algorithms in terms of their mechanism, SI algorithms leverage the simulated collective and social intelligence of these creatures to guide the interactions among search agents. The number of newly proposed SI algorithms exceeds those of EAs and PBs. One widely adopted SI algorithm is Ant Colony Optimization (ACO), which is utilized in several selected studies as a popular feature selection (FS) technique (RP1, RP3, RP5, RP7, RP12, RP15, RP16, RP20, RP26, RP34, RP42, RP52, RP59, RP61, RP63, and RP86). Additionally, Particle Swarm Optimization (PSO) is another notable SI algorithm employed in RP14, RP36, RP41, RP50, RP63, RP65, RP67, and RP100. It is worth emphasizing that the realm of Swarm Intelligence (SI) has witnessed a significant influx of novel algorithms in recent times, substantially broadening the array of choices available for optimization tasks. Among the more recent SI algorithms highlighted in the selected studies, notable examples include the Artificial Bee Colony (ABC) algorithm, applied in RP8, RP24, and RP88. Additionally, the Antlion Optimization (ALO) algorithm has been employed in RP9, RP18, and RP54, while the Crow Search Algorithm (CSA) has been identified in RP44, RP47, and RP77. Furthermore, the Whale Optimization Algorithm (WOA) has been utilized in RP30, RP55, RP76, and RP97. Other noteworthy SI algorithms encompass the Cuckoo Search (CS) algorithm, which has been employed in RP39, RP64, RP83, and RP105. Additionally, the Grasshopper Optimization Algorithm (GOA) has been found in RP31, RP56, RP60, RP66, and RP96, and the Grey Wolf Optimization (GWO) algorithm has been utilized in RP18, RP47, RP49, RP58, RP70, RP96, and RP101. and the Mayfly (MF) algorithm which is used in RP71 is introduced as a novel technique for addressing FS problems. This innovative method takes a hybrid approach, synergizing the strengths found in traditional optimization techniques like PSO, GA, and FA. These recent SI algorithms have substantially augmented the repertoire of options available for optimization tasks, providing researchers and practitioners with an expanded toolkit to effectively address intricate problems.

Furthermore, a diverse array of captivating swarm intelligence (SI) algorithms has emerged, significantly expanding the range of optimization techniques. These encompass the firefly algorithms (FFA) as scrutinized in RP23 and RP78, the Moth-Flame Optimizer (MFO) employed in RP18 and RP58, the Slap Swarm Optimization (SSO) explored in

RP35 and RP81, and the Brainstorm Optimization (BSO) utilized in RP59 and RP72. Additionally, the Bat Optimization Algorithm (BA) has been studied in RP38, RP90, and RP108, while the Water Wave Optimization (WWO) has been investigated in RP80. Noteworthy algorithms also include the Butterfly Optimization Algorithm (BOA) employed in RP48 and RP98, the Selfish Herd Optimizer (SHO) studied in RP75, the Social Spider Optimization (SSO) explored in RP57, and the Artificial Fish Swarm Algorithm (AFSA) examined in RP45. Moreover, the Shuffled Frog Leaping Algorithm (SFLA) has been discussed in RP51 and RP104, and the Symbiotic Organism Search (SOS) Algorithm has been utilized in RP53. Additionally, the Pigeon-inspired Optimization (PIO) algorithm has been studied in RP82, the Krill Herd Optimization (KHO) explored in RP84, and the Dandelion Algorithm (DA) utilized in RP91.

In addition, the Horse Herd Optimization Algorithm (HOA) was thoroughly investigated in RP92. RP94 employed the Dragonfly Algorithm (DA), which took inspiration from the natural behavior of dragonflies. These SI algorithms exemplify the ingenuity and diversity of drawing inspiration from various natural phenomena and collective behaviors. Each algorithm derives inspiration from distinct aspects of nature or collective behavior, with the shared goal of providing effective optimization solutions for diverse problem domains. By simulating the behavior of fireflies, moths, slaps, bats, water waves, butterflies, selfish herds, social spiders, artificial fish swarms, shuffled frogs, symbiotic organisms, pigeons, krill herds, dandelions, and horse herds, these algorithms strive to offer efficient optimization solutions for a wide spectrum of problem domains. The continuous advancement and exploration of such algorithms contribute to the ever-evolving field of optimization, offering promising avenues for addressing intricate optimization challenges across various domains.

Several metaheuristic (MH) techniques find inspiration from various mathematical theories and concepts. For instance, the Chaotic Optimization Algorithm (COA), applied in RP9, RP18, and RP44, draws influence from Chaos Theory. RP4 and RP11 utilize the Greedy Randomized Adaptive Search Procedure (GRASP), a metaheuristic algorithm tailored for solving combinatorial problems. GRASP involves the construction and local search phases in each iteration. The Binary Coordinate Ascent (BCA) algorithm, employed in RP10, takes inspiration from the well-known coordinate descent algorithm. Additionally, the Variable Neighborhood Search (VNS) technique (RP19, RP25) tackles global optimization and combinatorial optimization problems by modifying the neighborhood of the current solution during the search process in a systematic manner. RP79 employs the Iterated Greedy (IG) technique, which addresses challenging combinatorial optimization problems through two phases: destruction and construction. The Sine Cosine Algorithm (SCA), utilized in RP85 and RP87, emulates the behavior of sine and cosine functions to uncover optimal solutions for optimization problems. RP89 utilizes the Jaya Optimization Algorithm (JOA), which draws inspiration from the Sanskrit concept of "Jaya," signifying success or victory. This algorithm iteratively improves a population of solutions to discover an optimal or nearly optimal solution.

Table 6 presents a concise overview of the metaheuristic (MH) techniques utilized in the selected studies to address Research Question 1, focusing on Feature Selection (FS). These

**Table 6 Distribution of studies across MH techniques.**

| # | MH techniques | # of study | Studies ID | # | MH techniques | # of study | Studies ID |
|---|---|---|---|---|---|---|---|
| 1 | ACO | 15 | RP1, RP3, RP5, RP7, RP12, RP15, RP16, RP20, RP26, RP34, RP42, RP52, RP59, RP61, RP63 | 26 | EPD | 1 | RP31 |
| 2 | GA | 9 | RP13, RP22, RP27, RP32, RP33, RP37, RP40, RP46, RP62, RP106 | 27 | WCFS | 1 | RP43 |
| 3 | PSO | 8 | RP14, RP36, RP41, RP50, RP63, RP65, RP67, RP100 | 28 | AFSA | 1 | RP45 |
| 4 | GWO | 5 | RP18, RP47, RP49, RP58, RP70, RP96 | 29 | BOA | 1 | RP48 |
| 5 | GOA | 4 | RP31, RP56, RP60, RP66, RP96 | 30 | SFLA | 1 | RP51, RP104 |
| 6 | COA | 3 | RP9, RP18, RP44 | 31 | SOS | 1 | RP53 |
| 7 | SA | 3 | RP4, RP28, RP43 | 32 | SSO | 1 | RP57 |
| 8 | ALO | 3 | RP9, RP18, RP54 | 33 | BCA | 1 | RP10 |
| 9 | CSA | 3 | RP44, RP47, RP77 | 34 | ASO | 1 | RP73 |
| 10 | WOA | 3 | RP30, RP55, RP76, RP97 | 35 | ISA | 1 | RP74 |
| 11 | HS | 3 | RP2, RP6, RP13, RP71, RP99 | 36 | SHO | 1 | RP75 |
| 12 | GSA | 3 | RP21, RP28, RP68 | 37 | IG | 1 | RP79 |
| 13 | CS | 2 | RP39, RP64, RP83, RP105 | 38 | WWO | 1 | RP80 |
| 14 | FFA | 2 | RP23, RP78 | 39 | MVO | 1 | RP69 |
| 15 | ABC | 2 | RP8, RP24, RP88 | 40 | MF | 1 | RP71 |
| 16 | VNS | 2 | RP19, RP25 | 41 | PIO | 1 | RP82 |
| 17 | SSO | 2 | RP35, RP81 | 42 | KHO | 1 | RP84 |
| 18 | MFO | 2 | RP18, RP58 | 43 | JOA | 1 | RP89 |
| 19 | GRASP | 2 | RP4, RP11 | 44 | DA | 1 | RP91 |
| 20 | DE | 2 | RP8, RP55 | 45 | HOA | 1 | RP92 |
| 21 | SCA | 2 | RP85, RP87 | 46 | BHO | 1 | RP93 |
| 22 | BA | 1 | RP38, RP90, RP108 | 47 | DA | 1 | RP94 |
| 23 | BSO | 2 | RP59, RP72 | 48 | BCOA | 1 | RP95 |
| 24 | ICA | 1 | RP17, RP88 | 49 | GEO | 1 | RP102 |
| 25 | TLBO | 1 | RP29, RP103 | 50 | BBO | 1 | RP107 |

techniques have been utilized in the selected studies to tackle FS challenges and provide solutions in the context of Research Question 1.

## Statistical analysis

The statistical analysis plan has been developed to assess the significance of differences between MH models in Table 6, which involves calculating summary statistics such as odds ratios (OR) for meta-analysis. OR for the data provided in Table 6 were calculated by constructing a 2 × 2 contingency table for each MH technique compared to the reference group (ACO). The ACO MH technique was designated as the reference group because it was the most frequently used technique. Then, the odds of studies using each technique were compared to the odds of studies using the reference technique (ACO). The odds ratio

**Table 7 Odds ratios (OR) meta-analysis for MH-FS in text classification.**

| MH technique | Odds ratio (*vs.* ACO) | Interpretation |
| --- | --- | --- |
| GA | 9/15 = 0.6 | The odds of studies using GA compared to ACO are 0.6 times as likely. |
| PSO | 8/15 = 0.533 | The odds of studies using PSO compared to ACO are 0.533 times as likely. |
| GWO | 5/15 = 0.333 | The odds of studies using GWO compared to ACO are 0.333 times as likely. |
| GOA | 4/15 = 0.267 | The odds of studies using GOA compared to ACO are 0.267 times as likely. |
| COA | 3/15 = 0.2 | The odds of studies using COA compared to ACO are 0.2 times as likely. |
| SA | 3/15 = 0.2 | The odds of studies using SA compared to ACO are 0.2 times as likely. |
| ALO | 3/15 = 0.2 | The odds of studies using ALO compared to ACO are 0.2 times as likely. |
| CSA | 3/15 = 0.2 | The odds of studies using CSA compared to ACO are 0.2 times as likely. |
| WOA | 3/15 = 0.2 | The odds of studies using WOA compared to ACO are 0.2 times as likely. |
| HS | 3/15 = 0.2 | The odds of studies using HS compared to ACO are 0.2 times as likely. |
| GSA | 3/15 = 0.2 | The odds of studies using GSA compared to ACO are 0.2 times as likely. |
| CS | 2/15 = 0.133 | The odds of studies using CS compared to ACO are 0.133 times as likely. |
| FFA | 2/15 = 0.133 | The odds of studies using FFA compared to ACO are 0.133 times as likely. |
| ABC | 2/15 = 0.133 | The odds of studies using ABC compared to ACO are 0.133 times as likely. |
| VNS | 2/15 = 0.133 | The odds of studies using VNS compared to ACO are 0.133 times as likely. |
| SSO | 2/15 = 0.133 | The odds of studies using SSO compared to ACO are 0.133 times as likely. |
| MFO | 2/15 = 0.133 | The odds of studies using MFO compared to ACO are 0.133 times as likely. |
| GRASP | 2/15 = 0.133 | The odds of studies using GRASP compared to ACO are 0.133 times as likely. |
| DE | 2/15 = 0.133 | The odds of studies using DE compared to ACO are 0.133 times as likely. |
| SCA | 2/15 = 0.133 | The odds of studies using SCA compared to ACO are 0.133 times as likely. |
| BA | 1/15 = 0.067 | The odds of studies using BA compared to ACO are 0.067 times as likely. |
| BSO | 2/15 = 0.133 | The odds of studies using BSO compared to ACO are 0.133 times as likely. |
| WCFS | 1/15 = 0.067 | The odds of studies using WCFS compared to ACO are 0.067 times as likely. |
| AFSA | 1/15 = 0.067 | The odds of studies using AFSA compared to ACO are 0.067 times as likely. |
| BOA | 1/15 = 0.067 | The odds of studies using BOA compared to ACO are 0.067 times as likely. |
| SFLA | 1/15 = 0.067 | The odds of studies using SFLA compared to ACO are 0.067 times as likely. |
| SOS | 1/15 = 0.067 | The odds of studies using SOS compared to ACO are 0.067 times as likely. |
| GE | 1/15 = 0.067 | The odds of studies using GE compared to ACO are 0.067 times as likely. |
| PIO | 1/15 = 0.067 | The odds of studies using PIO compared to ACO are 0.067 times as likely. |
| KHO | 1/15 = 0.067 | The odds of studies using KHO compared to ACO are 0.067 times as likely. |
| JOA | 1/15 = 0.067 | The odds of studies using JOA compared to ACO are 0.067 times as likely. |
| DA | 1/15 = 0.067 | The odds of studies using DA compared to ACO are 0.067 times as likely. |
| HOA | 1/15 = 0.067 | The odds of studies using HOA compared to ACO are 0.067 times as likely. |

was calculated by dividing the number of studies using each technique by the number of studies using ACO as shown in Eq. (1). Then, the results were organized into Table 7.

$$Odds\ ratio\ (each\ technique\ vs.\ ACO) = \frac{Odds\ of\ studies\ with\ each\ technique}{Odds\ of\ studies\ with\ ACO} \tag{1}$$

**Table 8 Datasets used for MH-FS in text classification.**

| Text classification types | Datasets | Referred studies |
|---|---|---|
| Sentiment analysis text classification | IMDb movie and Initial tour medical blogs | RP15 |
| | Nikon, Nokia, Apex, Canon, and Creative from Amazon | RP20, RP42 |
| | OCA, Twitter, Political, and Software which is Arabic sentiment analysis datasets | RP55 |
| | Two Twitter benchmark corpora (SemEval 2016 and SemEval 2017) | RP58 |
| | Nine datasets that belong to four different contexts from Twitter social network | RP69 |
| | The Product Opinion Dataset from Amazon | RP71 |
| | Nine public sentiment analysis datasets (doctor, lawyer, drug, laptop, camera, radio, music, camp, and TV). four Amazon review datasets are DVD, electronics, books, and kitchen | RP79 |
| | The SemEval-2014 | RP83 |
| | Two datasets were gathered from Amazon1 reviews (Electronic and Movie), Sixteen UCI datasets (Iris, Heart, Hepatitis, Lung Cancer, Yelp, Lymph, Pima, Cancer, Diabetes, Heart-Stalog, Dermatology, Thyroid, Sonar, Gene, IMDB, and Amazon,), and three Twitter datasets (SOMD, STS-Test, and Sanders). | RP88 |
| Spam text classification | PU2, PU3, Lingspam, CSDMC2010, Trec2007, and Enron-spam. | RP2 |
| | WEBSPAM UK-2006 | RP36 |
| | Spam-Base dataset and Enron spam email *corpus* | RP43, RP92 |
| | Public shared *corpus* | RP84 |
| Text classification | Retures-21578 from UCI Repository | RP17, RP96 |
| | Alt, Structure, Disease, Function, Subcell, Acq, Money-fx, Corn, Earn, Ship, Grain, and Crude | RP19 |
| | Three open-source web applications (qaManager, bitWeaver, and WebCalendar) Two play store web application (Dineout: Table, Reserve, and Wynk Music) | RP41 |
| | OHSUMED | RP45 |
| | Twitter, ASKfm, and Formspring | RP64 |
| | Three public Arabic datasets, namely Akhbar-Alkhaleej, Alwatan, and Al-jazeera-News | RP49 |
| | OSAC which is collected from BBC and CNN Arabic websites | RP78 |
| | Turkish tweets obtained from three various GSM operators | RP37 |
| | Three Hadiths datasets | RP86 |
| | Chess and Email word subject | RP93 |
| | 20Newsgroups from UCI Repository | RP96, RP108 |
| | TR11WC and TR23WC | RP97 |
| | Society, Science, Reference, Recreation, Health, Entertainment, Enron, Education, Computer, Business, and Arts. | RP100 |
| | News Aggregator, News articles, RSS news feed, and Facebook news posts. | RP108 |

## RQ2: in the context of text classification, which specific MH techniques have been applied for FS?

In this section, the aim is to identify the metaheuristic-feature selection (MH-FS) techniques utilized in the selected studies for text classification. Additionally, we provide an overview of the commonly used datasets, classifiers, and performance evaluation metrics in the context of MH-FS for text classification machine learning techniques. As depicted in Table 5, a total of 28 studies (RP2, RP15, RP17, RP19, RP20, RP36, RP37, RP41,

RP42, RP43, RP45, RP49, RP55, RP58, RP64, RP69, RP78, RP79, RP83, RP84, RP87, RP88, RP92, RP93, RP96, RP97, RP100, and RP108) have employed MH techniques for FS in text classification. These studies offer insights into the application of MH-FS techniques, highlighting the datasets commonly used, the classifiers employed, and the frequently utilized performance evaluation metrics in the domain of text classification machine learning.

### RQ2.1: what are the datasets employed in the application of MH-FS for text classification?

Different datasets have been employed for MH-FS in text classification, depending on the specific classification task, such as sentiment analysis, spam classification, or general text classification. Additionally, the choice of datasets has been influenced by the language used, including English, Arabic, and Turkish. In Table 8, ten selected studies (RP15, RP20, RP42, RP55, RP58, RP69, RP79, RP83, and RP88) utilized sentiment analysis for text classification. All of these studies used the English language, except for RP55, which employed Arabic. Similarly, five studies (RP2, RP36, RP43, RP84, and RP92) focused on spam text classification. Among the studies that performed English text classification, RP17, RP19, RP41, RP45, RP64, RP49, RP78, RP37, RP93, RP96, RP97, RP100, and RP108 were identified, while RP49 and RP78 also incorporated Arabic text classification, and RP37 included Turkish text classification.

### RQ2.2: which classifiers have been used with MH-FS in text classification?

The text classification process encompasses three primary stages: text preprocessing, feature selection (FS), and constructing a text classification model with a machine learning classifier to evaluate the performance of different FS techniques. The selected studies have utilized different classifiers for FS-MH in text classification. Table 9 presents eight classifiers along with their definitions and the studies where they were applied for text classification. These classifiers include support vector machine (SVM), naïve Bayesian (NB), k-nearest neighbor (KNN), decision tree (DT), multilayer perceptron (MLP), artificial neural networks (ANN), centroid based algorithm (CBA), and AdaBoost. As shown in Table 9, SVM, NB, and KNN are the most commonly used classifiers in text classification, followed by DT and MLP, with AdaBoost being used to a lesser extent. The least utilized classifiers are ANN and CBA. Figure 5 provides a visual representation of the number of studies employing different classifiers in text classification.

### RQ2.3: which performance evaluation metrics are commonly utilized to assess the effectiveness of MH-FS in text classification?

Among the studies that have been reviewed in the field of MH-FS for text classification, a range of evaluation metrics have been utilized. These metrics serve to evaluate and compare the performance of diverse models developed through a variety of machine learning and statistical methods. The metrics in question include Precision, Recall, F-measure, Accuracy, AUC (area under the curve), the number of selected features, and

**Table 9 Classifiers used for MH-FS in text classification.**

| Classifiers | Definitions | Referred studies |
|---|---|---|
| SVM | SVM is the most successful supervised machine learning algorithm used for either classification or regression problems to determine the decision boundary between two classes to the maximum extent away from a point in the training dataset. By applying the kernel approach to transform the data, SVM may carry out either linear classification or non-linear classification and according to these transformations, it can determine an optimal boundary between the possible outputs (*Saraswathi & Tamilarasi, 2016*; *Alshalif et al., 2023*). | RP2, RP15, RP19 RP36, RP37, RP41, RP43, RP45, RP49, RP55, RP58, RP64, RP69, RP78, RP84, RP87, RP92, RP97 |
| NB | NB is among the popular practical supervised machine learning algorithms that is universally utilized especially for text classification and medical diagnosis because it is capable of scaling features of the large dimensions of space. NB is a simple probabilistic model according to the Bayes theorem. it depends on the hypothesis that attribute values are conditionally independent by looking at the targeted labels (*Chantar et al., 2020*; *Kumar & Jaiswal, 2019*; *Thiyagarajan & Shanthi, 2019*). | RP2, RP15, RP19 RP36, RP41, RP43, RP45, RP49, RP55, RP58, RP64, RP79, RP84, RP92. |
| KNN | KNN is the most basic and easiest supervised machine learning algorithm and is widely used in the text classification model. In the training data, it collects new data according to the shortest distance between k neighbors. The Euclidean distance formula is used to determine this distance. The basic concept of this classifier is that an object is classified based on the votes of the majority of its neighbors (*Ahmad, Bakar & Yaakub, 2019*; *Chantar et al., 2020*; *Kumar & Jaiswal, 2019*). | RP17, RP41, RP42, RP43, RP49, RP55, RP58, RP87, RP92, RP97 |
| DT | DT is a supervised machine-learning technique that resembles a tree and builds the classification tree using a set of training instances and it includes branches, root, and leaf nodes. Generally, the most widely used decision tree algorithm is C4.5, which is an improvement over the decision tree technique from the previous version (*Chantar et al., 2020*; *Kumar & Jaiswal, 2019*). | RP36, RP41, RP49, RP58, RP93, RP97 |
| MLP | MLP is a supervised machine learning algorithm it is one type of neural network. it includes three main layers which are input, hidden, and output layers. MLP is a self-adaptive and data-driven technique that can arrange them according to the data without explicitly defining a distributional or appropriate format for the underlying model (*Kumar & Jaiswal, 2019*). | RP36, RP41, RP58, RP92. |
| AdaBoost | AdaBoost is an appropriate algorithm for building a strong classifier from a combination of weak classifiers. it is considered to be suitable for real-time applications. Another advantage of AdaBoost it uses fewer features and less memory (*Thiyagarajan & Shanthi, 2019*). | RP36, RP45, RP83. |
| ANN | ANN is the organization and function of this model inspired by the biological neural networks of the human brain. it is a collection of interconnected processing units named neurons or nodes. It consists of five main elements which are inputs, weight, bias, activation function, and output. Each input is multiplied by weight to create the weighted inputs. the bias along with all weighted inputs are then added. Then, on the output neuron, an activation function is applying to the summary of prior weighted inputs and bias (*Dwivedi, 2018*; *Alshalif, Ibrahim & Waheeb, 2017*; *Alshalif, Ibrahim & Herawan, 2017*). | RP37, RP87. |
| CBA | CBA has been used to solve text classification issues. In this approach, the vector-space model is used to represent the documents. In the term space, each document is therefore viewed as a vector (*Ferrandin et al., 2015*). | RP37 |

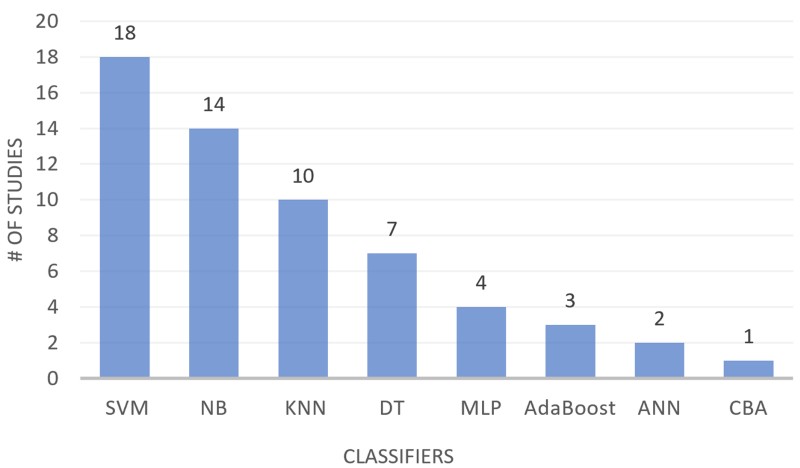

**Figure 5 Studies using different classifiers in text classification.**

| Table 10 Evaluation metrics used for MH-FS in text classification. | | |
|---|---|---|
| Evaluation metric | Definition | Referred studies |
| Precision | Precision is the positive predictive value and it is the ratio of documents to documents that are correctly categorized. By dividing the total number of true positives by the sum of true positives and false positives, it is calculated (*Mojaveriyan, Ebrahimpour-komleh & Jalaleddin, 2016*; *Gokalp, Tasci & Ugur, 2020*). | RP2, RP15, RP17, RP20, RP36, RP42, RP43, RP45, RP49, RP64, RP78, RP79, RP83, RP84, RP88, RP92, RP93, RP96 |
| Recall | Recall, often referred to as sensitivity, measures how well a model can identify all relevant instances within a dataset. It is measured as the ratio of true positives to both true positives and false negatives (*Mojaveriyan, Ebrahimpour-komleh & Jalaleddin, 2016*; *Gokalp, Tasci & Ugur, 2020*). | RP2, RP15, RP17, RP20, RP36, RP42, RP43, RP45, RP49, RP64, RP78, RP79, RP83, RP84, RP88, RP92, RP93, RP96 |
| F-measure | The weighted harmonic means of recall and precision, often known as the F-measure or F-score, is a metric for assessing the correctness of a test (*Singh & Kaur, 2020*; *Gokalp, Tasci & Ugur, 2020*; *Singh & Singh, 2018*). | RP2, RP15, RP17, RP20, RP36, RP42, RP43, RP45, RP49, RP64, RP78, RP79, RP83, RP84, RP88, RP96 |
| Accuracy | The accuracy rate (ACC) is the most general evaluation measure used in practice; it is used to evaluate classifier effectiveness according to the percentage of its correct predictions. Generally, it is determined by dividing the total number of true positives and true negatives by the total number of true positives, true negatives, false negatives, and false positives. A data point that the algorithm correctly identified as true or untrue is referred to as a true positive or true negative. Moreover, a data point that the algorithm misclassified as a false positive or false negative (*Tubishat et al., 2019*; *Al-Rawashdeh, Mamat & Hafhizah Binti Abd Rahim, 2019*). | RP15, RP19, RP37, RP41, RP43, RP55, RP58, RP69, RP79, RP84, RP87, RP88, RP92, RP93, RP96, RP97 |
| Number of selected features. | It is a parameter examined to measure the performance of the proposed FS technique. Low number of selected features indicated a better FS technique (*Senan et al., 2012*). | RP19, RP45, RP58, RP84. |
| AUC | AUC stands for Area under the ROC Curve, however, ROC stands for Receiver Operator Characteristic which is a probability curve that plots the True Positive Rate against False Positive Rate at different threshold values, while AUC determines degree or measure of separability it shows the capability of the model to distinguish between the positive and negative classes. Higher AUC shows a better performance of the model (*Narkhede, 2019*). | RP36, RP64, RP79, RP93, RP97 |
| Stability | The stability of a classification algorithm is the degree to which the same procedure may produce repeatable results when different batches of data are specified (*Turney, 1995*). | RP19. |

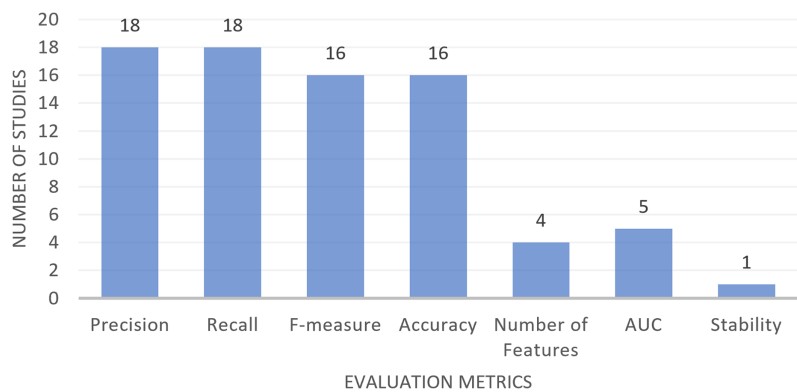

**Figure 6  Studies using different evaluation metrics in text classification.**

Stability. Table 10, detailed the descriptions and definitions of these evaluation metrics, along with information on which studies made use of them. Figure 6 has been prepared to offer a visual overview of the usage frequency of these metrics. As shown in Fig. 6, Precision and Recall are the most commonly employed metrics, with F-measure and Accuracy following closely behind. AUC and the count of selected features are also frequently employed for evaluation purposes. In contrast, Stability is a less commonly utilized metric in the analyzed studies.

## RQ3: is there empirical evidence indicating that MH-FS techniques outperform traditional FS methods in the domain of text classification?

According to the selected studies, the comparison was done based on two types of studies which are: (1) studies compared MH-FS and traditional FS techniques and (2) studies compared MH-FS techniques with existing MH-FS techniques. In this section, the focus is on comparing MH with the traditional one. The MH-FS techniques perform better than the traditional FS In this review, 26 articles extracted from the chosen studies (RP1, RP2, RP3, RP5, RP6, RP8, RP10, RP12, RP13, RP17, RP23, RP26, RP30, RP31, RP34, RP41, RP42, RP46, RP50, RP58, RP60, RP61, RP62, RP69, RP78, and RP79) undertake comparisons of their methodologies with various traditional FS techniques such as information gain, ReliefF, Laplacian score (L-score), Fisher score (F-score), relevance–redundancy feature selection (RRFS), random subspace method (RSM), Chi-square, mutual Information, symmetrical-uncertainty, minimal-redundancy–maximal-relevance (mRMR), sequential feature selection (SFS), correlation-based feature selection (CFS), and so many other techniques.

To provide a scientific explanation without the need to delve into detailed descriptions of all the techniques, two examples representing the mentioned technologies have been presented such as RP1 and RP2. These two studies were chosen to ensure a clear and comprehensive understanding for the reader without being overwhelmed with additional details. In RP1, the RRFS-ACO MH technique consistently outperformed traditional techniques, demonstrating superior results when compared to information gain, gain ratio, symmetrical-uncertainty, Gini index, F-score, term variance (TV), L-score, mRMR,

**Table 11 Traditional FS techniques.**

| Conventional methods | Definitions | Referred studies |
|---|---|---|
| Information gain | By assessing the gain in relation to the class, information gain determines the value of an attribute. Entropy, which is a measure of how chaotic or unpredictable a system is, is what it depends on to function. Information gain indicates the amount of knowledge left over after removing ambiguity (*Gokalp, Tasci & Ugur, 2020*). | RP2, RP8, RP17, RP30, RP31, RP34, RP42, RP50, RP60, RP78, RP79 |
| ReliefF | ReliefF is a single-variate, multiple-class, supervised, filter-based feature-weighting technique that can handle noisy and imperfect data. By periodically sampling an instance and seeing the value of the provided property for k of its closest examples belonging to the same and distinct classes, it determines the value of the attributes. Additionally, it uses a feature weighting scheme to investigate features with the greatest ability to distinguish between classes. with little computing complexity and unaffected by feature interactions (*Ghimatgar et al., 2018*; *Gokalp, Tasci & Ugur, 2020*). | RP5, RP12, RP26, RP46, RP50, RP60, RP69, RP79 |
| L-Score | The L-Score is a univariate method that evaluates each characteristic independently while ignoring interdependencies and relying on locality to preserve power. The local geometric structure of the data space is more significant in this approach than the overall structure. The nearest neighbor graph is used to model the local geometric structure (*Ghimatgar et al., 2018*). | RP1, RP3, RP5, RP12, RP13, RP26, RP61 |
| F-score | The F-Score is a univariate algorithm that operates solely on relevance analysis. To maintain power, the F-Score investigates variables with the greatest discrimination potential and the greatest number of locales. Similar to the L-Score, the F-Score evaluates features independently without taking into account how they are related (*Ghimatgar et al., 2018*). | RP5, RP12, RP26, RP30, RP31, RP50, RP60 |
| RRFS | A multivariate approach called RRFS evaluates features based on their maximal relevance to classes and their minimal redundancy with respect to one another. Mean Absolute Difference (MAD) or Mutual Information (MI) are two supervised or unsupervised relevance criteria that can be employed in this algorithm to assess the significance of each feature (*Ghimatgar et al., 2018*). | RP1, RP3, RP5, RP13, RP26, RP61 |
| Correlation-based Feature Selection (CFS) | CFS is an algorithm that ranks a subset of features based on a heuristic evaluation function that relies on correlations. This algorithm starts to build a correlation matrix between the features in the dataset. Then, a search metaheuristic is utilized to build subsets of features to be ranked. The rank assigned to the subsets that result is the correlation between the features and the class divided by the intercorrelation of the features between themselves (*Salama, Abdelbar & Anwar, 2016*). | RP8, RP30, RP31, RP50, RP60 |
| Chi-square | Chi-square also known as χ2 Statistic is determined by computing the value of the chi-squared statistic in relation to the class, it determines the worth of an attribute. It is compared to the χ2 distribution with one degree of freedom in order to assess the lack of independence between terms and classes (*Gokalp, Tasci & Ugur, 2020*). | RP2, RP8, RP46, RP79 |
| RSM | To better manage the noise in high-dimensional datasets, the Random Subspace Method (RSM) applied a multivariate search methodology to a randomly chosen subset of features (*Tabakhi & Moradi, 2015*). | RP1, RP3, RP13, RP61 |
| TF-IDF | It is a numerical statistical method for determining the significance of a term for a set of documents (*Saraswathi & Tamilarasi, 2016*). | RP2, RP41, RP58, RP78 |
| mRMR | A multivariate method called mRMR uses assessments of redundancy and relevance. It evaluates a feature subset with the least amount of overlap between features and the most amount of class relevance. Average F-statistic values are calculated over various time steps to execute the relevance analysis. The dynamic time-warping method employs the redundancy analysis (*Ghimatgar et al., 2018*). | RP5, RP12, RP26 |

| Table 11 (continued) | | |
|---|---|---|
| Conventional methods | Definitions | Referred studies |
| Sequential Feature Selection (SFS) | SFS is a wrapper algorithm that begins with an empty set and iteratively tries to add features that maximize the current predictive accuracy of the learning algorithms (*Salama, Abdelbar & Anwar, 2016*). | RP10, RP23, RP62 |
| Mutual information | It is an information theory basic concept. It is a measurement of general interdependence among two random variables (*Tourassi et al., 2001*). | RP17, RP62 |
| symmetrical-uncertainty | Symmetrical-uncertainty coefficient is an improved version of information gain that minimizes the bias across the multivalued features. It assesses the value of an attribute by measuring the symmetrical uncertainty with regard to the class (*Gokalp, Tasci & Ugur, 2020*). | RP46, RP79 |

mutual correlation (MC), RSM, and RRFS. In RP2, the global best harmony-oriented harmony search (GBHS) achieved superior results when compared to conventional methods, including Chi-square, feature selection based on comprehensive measures, t-test-based feature selection, information gain using term frequency, and an improved term frequency-inverse document frequency approach. Table 11 lists the most frequently used traditional FS techniques along with their definitions and their studies.

## RQ4: what are the discernible strengths and weaknesses of MH techniques in the context of FS?

This section identifies and summarizes the strengths and weaknesses of metaheuristic (MH) techniques as reported by researchers. it is focusing on the strengths and weaknesses that have been supported by multiple studies. MH techniques have demonstrated strong performance in feature selection (FS) problems. They are particularly praised for their ability to effectively handle redundant features and high-dimensional data. Table 12 summarizes the strengths of MH techniques based on the selected studies, along with the studies that support each strength. Concurrently, Table 13 summarizes the weaknesses of MH techniques, accompanied by the studies that provide evidence for each weakness. In summary, it is important to note that different MH techniques have varying advantages, and there is no universal solution for MH-FS techniques that fits all scenarios.

According to the previous discussion it expected that some different solutions that can efficiently improve the MH-FS which are as follows: use MH with the binary system, integrate with traditional FS techniques, integrate with some mathematics theories such as Chaos theory and Rough Set theory, and combine two MH techniques to take benefit from their advantages to complement each other. In addition, there are several reasons which inspire FS techniques to adopt MH techniques as highlighted by the research articles in RP3, RP4, RP6, RP42, RP53, RP57, and RP66. The reasons that motivate researchers to adopt MH in FS are summarized in Fig. 7. Briefly, there are various aspects of MH for FS following section highlights their strengths and discusses each aspect.

**Table 12 Strength of MH techniques.**

| Technique | Strengths | Studies |
|---|---|---|
| ACO | • Positive feedback that leads to find rapid and good solutions.<br>• Easy and nature implementation in a parallel way.<br>• low execution time in optimization.<br>• Can increase the local and global search capabilities due to the greedy and stochastic natures of the algorithm.<br>• Used of distributed long-term memory.<br>• Used the same structure of reinforcement learning schema<br>• A population-based algorithm that used the colony of ants which leads to raising the robustness of this algorithm.<br>• Efficient and competent in the convergence process. | RP1, RP3, RP5, RP12, RP26, RP42 RP61 |
| Advanced binary ACO | • ABACO allows ants to search for all features.<br>• In the ABACO algorithm ants are authorize to select or deselect visiting features.<br>• ABACO is not constrained to preselect or deselect specific features.<br>• ABACO incorporates heuristic desirability to enhance the exploration of the search process and guide ants towards more prominent features. | RP7 |
| HS | • Free from divergence.<br>• No initial value settings of the decision variables are required.<br>• The algorithm has the capability to identify and select the most suitable individuals, ensuring that their optimal harmonies are preserved and carried forward to subsequent iterations.<br>• Parallel exploration of the search space for the given data. | RP2, RP6 RP13 |
| PSO | • Easy to implement.<br>• Fast convergence speed.<br>• Global communication between the particles.<br>• Able to produce quick solutions for nonlinear optimization problems. | RP36, RP54, RP81 |
| ABC | • Easy implementation.<br>• Demonstrates significant robustness.<br>• Exhibits high flexibility.<br>• Requires fewer control parameters.<br>• Excels in exploitation during the onlooker bee processing phase. | RP8 |
| DE | • Requires a smaller number of parameters.<br>• Operates at a high speed.<br>• Exhibits robustness.<br>• Suitable for tackling high-dimensional and complex optimization problems. | RP8 |
| GA | • Able to search solution space in combinatorial optimization problems.<br>• Able to solve the nonlinear and complex problems. | RP40, RP54, |
| CS | • Less parameters to be tuned.<br>• Can adapt to a wider class of optimization problems.<br>• Fast convergence.<br>• Global optima achievement. | RP39 |

| Table 12 (continued) | | |
|---|---|---|
| Technique | Strengths | Studies |
| ALO | • ALO has the ability to deliver very competitive and promising performance. | RP54 |
| Chaotic (CALO) | • CALO demonstrates the capability to converge towards the same optimal solution across a wide range of applications. | RP9 |
| CSA | • When addressing complex, high-dimensional, and multimodal problems, it is straightforward to avoid local optima. | RP47 |
| SSO | • Utilizes basic mathematical operators to discover the optimal solution.<br>• Offers a cost-effective approach in terms of time complexity and space complexity. | RP57 |
| BSO | • Use of clustering in the iterative process to create an optimization algorithm. | RP72 |
| SA | • Can escape the local minimums.<br>• In each iteration, it requires a single evaluation of the wrapper.<br>• Enables control over the trade-off between solution length and result precision. | RP4 |
| TLBO | • Needs only the general control parameters to be tuned. | RP29 |
| WCA | • Characterized by a minimal number of control parameters (only 3).<br>• Effectively tackles the issue of rapid convergence towards local optima entrapment through the implementation of an evaporation technique. | RP43 |
| GWO | • Simple and easy to use.<br>• Fast convergence.<br>• Adaptable and capable of scaling to different contexts or sizes.<br>• Few parameters to tune.<br>• Demonstrates a certain degree of capability to prevent stagnation in local optima.<br>• Achieves a favorable balance between exploration and exploitation through a straightforward approach.<br>• Inspired by the intelligent leadership and hunting behaviors observed in grey wolves in nature. | RP47, RP49, RP70 |
| SFLA | • Characterized by a straightforward structure.<br>• Involves a reduced number of controlling parameters.<br>• Features a simple implementation of the algorithm. | RP51 |
| SSA | • Requires a reduced number of parameters.<br>• Straightforward to implement.<br>• Ability to solve large-and-small-scale problems.<br>• Flexible and strong stochastic nature. | RP81 |
| AFSA | • Demonstrates resilience and robustness.<br>• Straightforward and easy to comprehend.<br>• Prone to being influenced by initial values due to its dependent on heuristic global optimization.<br>• Great influence by the fish behavior in the convergence speed and global search. | RP45 |

| Technique | Strengths | Studies |
|---|---|---|
| BOA | • Provides a comprehensive explanation of competitive results by considering factors such as exploration, exploitation, convergence, and avoidance of local optima.<br>• Demonstrates strong performance across a wide range of unimodal and multimodal benchmark functions.<br>• High convergence rate.<br>• Assists in achieving high exploration by utilizing fragrance attenuation, which facilitates an efficient search across the solution space. | RP48 |
| SOS | • Simple structure.<br>• Straightforward implementation.<br>• No parameter requirements.<br>• Exhibits remarkable stability.<br>• Rapid convergence speed.<br>• Produces high-precision solutions.<br>• Avoids getting trapped in locally optimal solutions.<br>• The mutualism and commensalism phases facilitate the population's quick focus on the vicinity of the optimal solution. | RP53 |
| WOA | • Demonstrates the ability to achieve a balance between exploration and exploitation. | RP55 |

**Table 13 Weaknesses of MH techniques.**

| Technique | Weaknesses | Studies |
|---|---|---|
| ACO | • Limited efficiency when dealing with datasets containing a large number of features.<br>• Slow convergence speed.<br>• Time-consuming execution.<br>• High space complexity leading to premature convergence.<br>• Computationally expensive in terms of memory requirement and speed.<br>• Computational complexity<br>• The efficiency of this algorithm is strongly influenced by the size of the selected feature subsets. | RP1, RP5 RP26, RP39, RP42 |
| HS | • The primary limitation of HS is the excessive number of iterations required to find an optimal solution. | RP2 |
| PSO | • Slow convergence speed.<br>• Time-intensive.<br>• Elevated space complexity and premature convergence.<br>• Inefficient trade-off between local and global search performance.<br>• Exhibits a weakness in fine-tuning near locally optimal positions.<br>• Not as effective when applied to large-scale problems. | RP39, RP50, RP81, RP93 |
| ABC | • The algorithm is time-consuming for convergence, and it fails to showcase its true performance adequately. | RP8. |
| DE | • Exhibits unstable convergence.<br>• Proneness to getting stuck in local optima. | RP8 |

| Technique | Weaknesses | Studies |
|---|---|---|
| GA | • The GA algorithm has a limitation related to the crossover operator, which can cause sudden and drastic changes to the solutions during the search process.<br>• The GA can result in slow convergence due to its lack of guidance, hindering effective exploration of the search space.<br>• GA requires the tuning of multiple parameters. | RP81, RP93. |
| CS | • Requires much time to compute the fitness function.<br>• slow convergence rate. | RP39 |
| ALO | • Select Sub-optimal selection due to unbounded random walk in the search space.<br>• Stagnation because the exploration capability is very limited.<br>• Local optima and premature convergence problems. | RP9, RP81 |
| CALO | • The optimization results may not be exactly repeatable.<br>• Careful consideration is required when transitioning to a different classifier, particularly in real-time applications.<br>• Increasing the running time when switching to another classifier. | RP9 |
| CSA | • Inefficient local search strategy.<br>• Low convergence rate due to trap in local optima.<br>• The stochastic nature of CSA introduces ambiguity in distinguishing between exploitation and exploration, leading to an unclear boundary between the two. | RP44, RP47, RP77 |
| SSO | • Explores the search space predominantly in one direction.<br>• Lacks information about other regions of the search space.<br>• Fall into sub-optimal solutions that affect algorithm performance. | RP57 |
| BSO | • The shortage of exploration relies directly on the algorithm internal configuration.<br>• Configuring the control parameters of BSO is a challenging task.<br>• The process of exploitation is influenced by the method used to create clusters. | RP72 |
| WCA | • The effectiveness of the algorithm as a spam classifier remains uncertain or ambiguous. | RP43 |
| GWO | • Cannot always perform exploration well.<br>• The algorithm is not always capable of successfully addressing the problem and may fail to find the global optimal solution. | RP47 |
| SFLA | • Negative affected on convergence speed and solving precision. | RP51 |
| SSA | • The algorithm is susceptible to issues related to population diversity and can become trapped in locally optimal solutions. | RP81 |
| FFA | • Waste of computation resources.<br>• Exhibits low efficiency in searching for optimal regions.<br>• Requires the tuning of numerous control parameters. | RP23 |

(Continued)

| Table 13 (continued) | | |
|---|---|---|
| **Technique** | **Weaknesses** | **Studies** |
| GSA | • Experiences premature convergence as a result of rapid deduction diversity.<br>• Demonstrates fast initial convergence during the early stages of the search process, which gradually slows down as the global solution approaches the local solution.<br>• Challenges arise in achieving a proper balance between exploration and exploitation. | RP28, RP68 |
| SHO | • Prone to getting trapped in local optimal solutions.<br>• Demonstrates low precision.<br>• Exhibits slow convergence speeds. | RP75 |

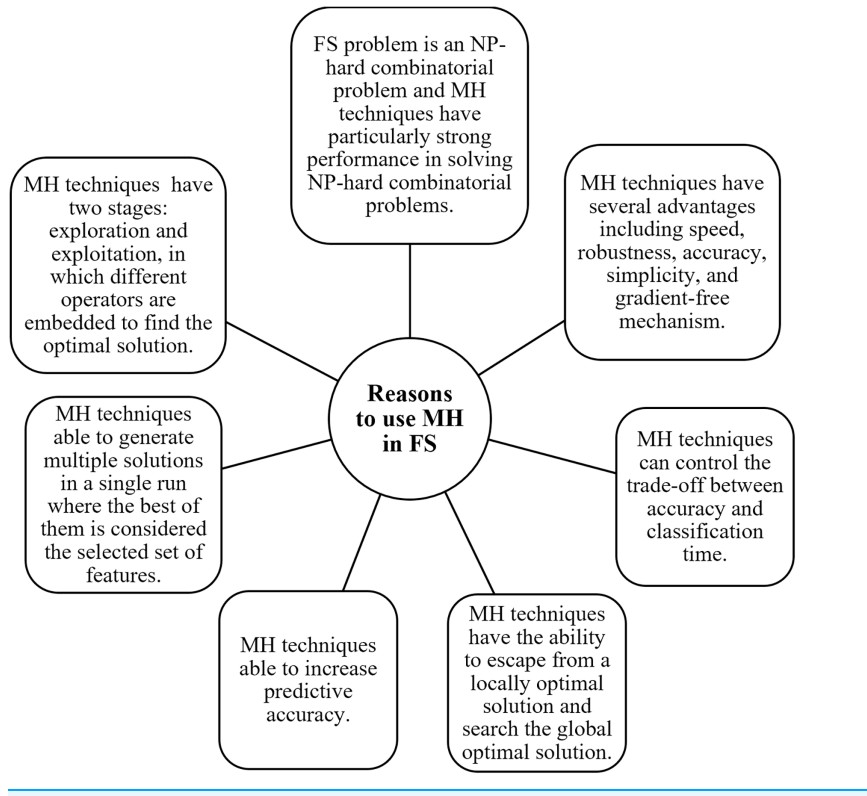

**Figure 7** Reasons to use MH in FS.

## Aspects of MH based on FS

The various strengths of MH approaches for feature selection, including their robust search strategies, global optimization capabilities, parallelizability, adaptability, innovation potential, and cross-domain applicability, make them a powerful choice for addressing feature selection challenges across a wide range of applications. Researchers and practitioners often turn to MH when seeking efficient and effective solutions for feature subset selection.

1) Search strategy: MH approaches are known for their powerful search strategies. They excel at navigating complex, high-dimensional search spaces to find the optimal subset of features. These strategies balance exploration and exploitation to efficiently explore the space while exploiting promising solutions. This dynamic search process is a significant strength of MH, enabling them to find feature subsets that traditional methods might overlook due to their deterministic nature.

2) Global optimization: One of the key strengths of MH approaches is their ability to perform global optimization. In FS, it is crucial to find the best combination of features that leads to optimal model performance. MH can efficiently search for solutions across the entire feature space, which traditional methods may struggle to do. This global exploration capability is particularly valuable when dealing with large and complex feature sets.

3) Parallelization and scalability: MH approaches often lend themselves well to parallelization. This means they can be distributed across multiple processors or machines to expedite the search process, making them scalable for large datasets and high-dimensional feature spaces. This scalability is crucial for handling big data and complex applications, further showcasing the practicality of MH feature selection.

4) Adaptability: Another advantage of MH is its adaptability to various optimization objectives and constraints. Researchers can easily customize the objective function to reflect the specific goals of FS problem, such as maximizing classification accuracy, minimizing model complexity, or considering trade-offs.

5) Exploration of new techniques: MH approaches encourage the development and exploration of new techniques. Researchers continually innovate and propose novel metaheuristic algorithms specifically designed for FS. For example, the RSS may represent such an innovation. This innovation-driven aspect of MH contributes to the field's dynamism and evolution.

6) Cross-domain applicability: MH approaches are not limited to a single domain. They can be applied to FS in various fields, not just text classification. This cross-domain applicability demonstrates the versatility and effectiveness of these techniques in solving FS problems in diverse application areas.

### RQ5: how can the RSS be effectively leveraged as an FS technique?

RSS, which stands for Ringed Seal Search, is a metaheuristic (MH) technique introduced in reference (*Saadi et al., 2016*). It draws inspiration from the natural behavior of seal pups when selecting a secure lair to evade predators. In comparison to other MH techniques like genetic algorithms (GA) and particle swarm optimization (PSO), RSS demonstrates faster convergence towards the global optimum and maintains a better trade-off between exploration and exploitation (*Saadi et al., 2016*). Although RSS has not been widely employed as a feature selection (FS) technique according to the existing literature, it possesses the capability to optimize SVM parameters. Consequently, this optimization leads to enhanced classification accuracy when compared to traditional SVM approaches (*Sharif et al., 2019*). The RSS algorithm is primarily inspired by the search behavior of seal

pups seeking optimal lairs to evade predators. It adopts a similar approach where the algorithm continually searches for better solutions and moves towards them. In the context of RSS, these "lairs" correspond to problem-specific representations, and the algorithm aims to optimize the quality or fitness of these representations. By iteratively improving the representations, RSS strives to identify the best possible solution. The scenario begins once the female seal gives birth to a pup in a birthing lair that has been created for this purpose. The seal pup technique entails of searching for and choosing the ideal lair by conducting a randomized walk to discover a new lair. The seal pup's random walk alternates between the normal and urgent search modes because seals are sensitive to outer noise produced by predators. The pup's normal mode is an intensive search among closely spaced lairs, which is described by Brownian walk. In urgent state, the pup leaves the proximity area and implements extensive search to discover new lair from scatter targets; this movement is described as Levy walk. The change among these two modes is realized by the random noise released *via* predators. The algorithm holds changing between normal and urgent modes until the global optimum is reached.

RSS is especially based on seal pup search for optimal lairs to escape predators. Each time a new lair that has perfect quality is found; the pup moves into it. In the end, the lair (habitat) with the optimal fitness (quality) it is the term that RSS is going to optimize. The RSS concepts is represented on the following depictions.

i) Each female seal gives birth to a single pup at a time, selecting a random habitat for the pup.

ii) The seal pup randomly explores its ecosystem to locate a suitable lair for protection against predators.

iii) The movement of the seal pup can be categorized into two states: Normal, where the search is focused and follows a Brownian walk, and Urgent, where the search is expansive and follows a Levy walk.

iv) If the best-seen lair $L^{best,t}$ among the current set of lairs $K$ is superior in terms of fitness value compared to the best lair $L^{best,t-1}$ from the previous iteration, $L^{best}$ is updated to $L^{best,t-1}$. Otherwise, $L^{best}$ remains unchanged.

Over time, inferior lairs will be discarded, and the seals will continue to explore and move towards better lairs or chambers, leading to convergence towards good solutions. The RSS algorithm will be adapted for feature selection by using the following steps:

1. Input: Provide the initial number of lairs for the search.

2. Output: The algorithm aims to find the best lairs based on some evaluation criteria.

3. Initialization: Generate birthing lairs' initial number: Initialize the lairs as L_1 = (f = 1, 2, 3,…, *n*), where *n* is the initial number of lairs.

4. While (Stopping criterion): Repeat the following steps until a specific stopping criterion is met. This criterion could be a maximum number of iterations, reaching a satisfactory solution, or other conditions specific to the problem being solved.

Algorithm: Ringed Seal Search (RSS) Feature Selection Algorithm
**Begin**

    Input: Initial Ranked Terms
    Output: Best Features
    Purpose= To select the best subset of features

  Generate the birthing lairs' initial number,
$$L_1 = (f = 1,2,3, \ldots \ldots \ldots, n)$$
**While** (Stopping criterion)

    **If** $noise = false$
      Search in the proximity for a new lair using Brownian walk;
    **Else**
      Expand the search for a way for a new layer using levy walk;
    **End if**
      Evaluate the fitness of every new lair using AUC and compare with previous;

    **If** $L^{best,T} > L^{best,T-1}$

      Choose the new lair
$$L^{best} = L^{best,T}$$
      Go to 4

    **End if**

  Select the lair;
  Return the best lair
  The global best lair is fed to classifiers for training.
**End**

**Figure 8** RSS feature selection algorithm. 

5. If noise = false: Check if the noise parameter is set to false. If so, perform a Brownian walk in the proximity to search for a new lair. A Brownian walk is a random process where the next step is determined by a random direction.

6. Else: If the noise parameter is set to true, expand the search for a new lair using a Levy walk. A Levy walk is a random process that allows for long-range exploration and can provide more global search capabilities.

7. Evaluation: Evaluate the fitness of every new lair and compare with previous: Assess the fitness of each newly generated lair using an appropriate evaluation function metrics such as accuracy, recall, precision, *etc*. Compare the fitness of these lairs with the previously evaluated lairs.

8. If $L^{\wedge}(best, T) > L^{\wedge}(best, T - 1)$ : Check if the fitness of the current best lair ($(L^{\wedge}(best, T)$ is greater than the fitness of the previous best lair ($L^{\wedge}(best, T - 1)$).

9. Choose the new lair: If the fitness of the current best lair is greater, select it as the new best lair ($L^{\wedge}best = L^{\wedge}(best, T)$).

10. Else: If the fitness of the current best lair is not greater, go to step 4 and continue the search.

11. Rank the lairs: Once the stopping criterion is met, rank the lairs based on their fitness evaluations.

12. End the loop.

13. Termination: Return the best feature subset found as the result of the feature selection.

RSS can be employed for feature selection by following the previous steps. Firstly, the problem should be defined, and the features must be represented appropriately. Secondly, a population of potential feature subsets is initialized, and their fitness is evaluated using a suitable metric. Thirdly, the RSS algorithm is applied iteratively, with the search space being explored, the fitness function being assessed, and the best solutions being exploited. A termination criterion is defined to determine when the algorithm should stop. Finally, the selected features can be extracted from the best solution after the algorithm finishes. Figure 8 summarizes the RSS feature selection algorithm.

## Future research directions

In this SLR, future research direction could focus on the further refinement and development of existing MH techniques and investigate how this technique can be optimized to enhance their performance in feature selection. It was noted that there are still some MH techniques, such as RSS, which have not been extensively explored for FS despite their effectiveness. RSS employs two search states, normal and urgent, and dynamically switches between them until the optimal solution is reached. This balance between exploitation and exploration enables RSS to find global optima faster than other techniques. Moreover, RSS has shown high accuracy in text classification problems, making it a promising choice for FS. Therefore, as a future direction, the study suggests the utilization of RSS as an FS technique in this research.

Further considerations addressing the challenges in feature selection research based on MH techniques require a multifaceted approach. Researchers can develop novel algorithms suitable for handling large-scale datasets efficiently while maintaining robustness and scalability. Additionally, strategies for handling dynamic environments and evolving datasets must be devised to ensure the adaptability of feature selection methods over time. Improving the interpretability and explainability of MH-based feature selection models is essential for gaining insights into the decision-making process. Hybrid approaches combining MH with other optimization techniques, machine learning algorithms, binary systems, or mathematical theories such as chaos theory can leverage the strengths of each method, leading to enhanced performance and flexibility. Additionally, the limitations of MH techniques require targeted strategies tailored to each algorithm's weaknesses. For instance, to overcome the challenges associated with ACO, efforts can focus on developing enhanced variants that improve efficiency and convergence speed, possibly through parameter tuning or hybridization with other optimization methods. Similarly, for HS, PSO, and ABC, optimizations could target convergence speed and space complexity by refining the search strategies or introducing adaptive mechanisms. DE's instability and GA's issues with crossover could be mitigated by incorporating diversity maintenance mechanisms or alternative operators. In addition, for algorithms like CSA and SSO, refining the balance between exploration and exploitation is crucial, possibly through algorithmic modifications or parameter adjustments. Finally, interdisciplinary research can uncover applications beyond text classification, while comparing MH techniques with deep learning methods can offer insights into their relative strengths and limitations.

## CONCLUSIONS

This study conducted a comprehensive SLR to assess the efficiency and effectiveness of MH techniques in the context of FS. The review rigorously followed a structured approach, encompassing the identification and quality assessment of 108 primary studies conducted between 2015 and 2022. These studies' characteristics were summarized based on the defined research questions, revealing that MH techniques have seen widespread adoption for FS across diverse domains, notably in text classification. Comparative analysis of MH techniques against traditional methods demonstrated their substantial enhancements in the performance of machine learning techniques, specifically within the field of classification. The strengths and weaknesses of MH techniques were meticulously scrutinized, with insights drawn exclusively from the selected studies. Moreover, this research unveils a promising avenue for future investigations, particularly emphasizing the potential for further exploration of MH techniques, as exemplified by the RSS, to refine feature selection across various application domains. This research significantly contributes to our comprehension of the central role MH techniques play in the realm of FS and their broader implications for the fields of data science and text classification. In summary, the findings underscore the compelling case for the adoption of MH techniques in feature selection, emphasizing their superior performance in text classification and serving as a catalyst for ongoing innovation and advancement in this crucial domain.

### Funding

Funding was provided by the Research Management Center at Universiti Teknologi Malaysia (Vot No: Q.J130000.21A6.00P48) and the Data Analytics and Artificial Intelligence (DAAI) Research Group in Birmingham City University, UK. The funders had no role in study design, data collection and analysis, decision to publish, or preparation of the manuscript.

### Grant Disclosures

The following grant information was disclosed by the authors:
Research Management Center at Universiti Teknologi Malaysia: Q.J130000.21A6.00P48.
Data Analytics and Artificial Intelligence (DAAI) Research Group in Birmingham City University, UK.

### Competing Interests

Faisal Saeed is an Academic Editor for PeerJ.

### Author Contributions

- Sarah Abdulkarem Al-shalif conceived and designed the experiments, performed the experiments, analyzed the data, performed the computation work, prepared figures and/or tables, authored or reviewed drafts of the article, and approved the final draft.

- Norhalina Senan conceived and designed the experiments, prepared figures and/or tables, and approved the final draft.
- Faisal Saeed performed the experiments, authored or reviewed drafts of the article, and approved the final draft.
- Wad Ghaban analyzed the data, prepared figures and/or tables, and approved the final draft.
- Noraini Ibrahim analyzed the data, authored or reviewed drafts of the article, and approved the final draft.
- Muhammad Aamir performed the computation work, authored or reviewed drafts of the article, and approved the final draft.
- Wareesa Sharif analyzed the data, authored or reviewed drafts of the article, and approved the final draft.

## Data Availability

This is a literature review.

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
