# Peer review of "A systematic literature review on meta-heuristic based feature selection techniques for text classification"

_PeerJ Computer Science, doi:10.7717/peerj-cs.2084_

## Round 0.1 · original submission · Major Revisions

Dear authors,

Thank you for submittinb your review article. Reviewers have not commented on your article and suggest major revisions. When submitting the review, specially it will be better to address followings:

1- Please include future research directions.
2- Present strong discussions leading to the different aspects of metaheuristic approaches based feature selection
3- Include explicit research questions guiding the review early in the paper.
4- Mention the comparison metrics.
5- Include indication of how you addressed or accounted for biases during the review.
6- A broader and more transparent methodology, careful database selection, and a more fair review of the techniques might all help the study. While the findings are intriguing, it requires thorough support and should be presented in a clear manner that systematically breaks down the complexities of the topic.

Reviewer 1 ·

Basic reporting

Yes, the topic of feature selection is of cross-disciplinary interest. Beyond computer science, fields like linguistics, digital humanities, business analytics, and more can benefit from advancements in text classification. Meta-heuristic methods and their applications are of interest to a wide variety of researchers working on optimization problems across various domains. The topic does seem to be within the scope of the journal as it deals with computer science aspects of data processing and machine learning. The application of meta-heuristic techniques to feature selection in text classification is an algorithmic and methodological topic that fits well within the realms of computer science research.

Experimental design

The authors have undertaken a systematic literature review, focusing on meta-heuristic based feature selection techniques for text classification from 2015 to 2022. They sourced from three databases: Scopus, Science Direct, and Google Scholar, summarizing 108 primary studies. Here are some aspects to consider when evaluating the survey methodology for comprehensive and unbiased coverage: The authors used three well-known databases. However, while Scopus and Science Direct are reputable academic databases, Google Scholar can sometimes include non-peer-reviewed or gray literature. It's important to understand how the authors ensured quality when extracting data from Google Scholar. The absence of databases like IEEE Xplore or PubMed, depending on the topics covered, may have resulted in missed studies. The authors chose studies from 2015 to 2022. While this seven-year window is quite recent and may capture contemporary techniques, it might exclude foundational or seminal works from before 2015 that influenced the field.

Validity of the findings

Yes, the conclusion does identify unresolved questions, gaps, and future directions. The authors noted that certain meta-heuristic (MH) techniques, like the Ringed Seal Search (RSS), have not been extensively explored for feature selection despite their demonstrated effectiveness. This indicates a gap in the current literature or application of this particular method. The study suggests a specific future direction, emphasizing the utilization of the Ringed Seal Search (RSS) as a feature selection technique. The authors provide a rationale for this recommendation by mentioning the capability of RSS to dynamically switch between search states and its potential for higher accuracy in text classification problems. The authors acknowledge a limitation in their systematic review, pointing out that the findings on the strengths and weaknesses of MH methods were derived solely from the selected studies. They highlight the potential unreliability of some findings, as they may be based on the subjective opinions of the respective original authors. So, the conclusion does touch upon unresolved questions, suggests future research directions, and acknowledges limitations in their review.

Additional comments

The study overtly suggests that MH techniques consistently outperform traditional methods without providing a granular breakdown of where and how. A claim of this magnitude requires substantial empirical support, and the study should have stratified its findings according to different scenarios or data types. The paper mentions comparing the performance of MH techniques to traditional methods but does not delve into the specifics of the comparison metrics. This absence makes it hard to gauge the true impact and novelty of the findings.
The contribution of the current review study over previous related reviews (such as Abu Khurma et al. A Review of the Modification Strategies of the Nature Inspired Algorithms for Feature Selection Problem. Mathematics. 2022; 10(3):464) should be clearly stated. While discussing underexplored techniques like RSS is important, the paper seems to place undue emphasis on its merits without providing comparative metrics or case studies. It is critical to justify why a particular technique, like RSS, is more advantageous over others.
For a systematic review, the research questions guiding the review should be explicitly outlined early in the paper. This provides readers clarity on what the paper aims to achieve.
The authors rightly note a limitation about the subjective opinions derived from the selected studies, but there's no indication of how they addressed or accounted for these biases during their review. If the study merely aggregates these subjective opinions without critical evaluation, the findings may be misleading.
In summary, while the authors have embarked on a pertinent topic in the realm of text classification, the study could benefit from a broader and more transparent methodology, careful selection of databases, and a more balanced analysis of the techniques. Their findings, while interesting, need rigorous backing and should be presented with a clear structure that systematically breaks down the nuances of the topic.

Reviewer 2 ·

Basic reporting

1. Organization of the content is good

Experimental design

Different aspects considered for reviewing the literature

Validity of the findings

Tabulations and figures provide a broader view of the different papers

Additional comments

No strong discussions leading to the different aspects of MH feature selection techniques are given. It lacks detailed explanation.
Future directions for research is missing

---

## Round 0.2 · Minor Revisions

Please consider the final suggestions by reviewers 3 (statistical testing) and 4 (fleshing out future research directions) in your revision.

While reviewer 1 has provided a negative review complaining about the lack of rigour resulting from not using the PRISMA methodology, I can see that a PRISMA flow chart has been provided as supplementary material so it seems that they might have been looking at an earlier revision.

Reviewer 1 ·

Basic reporting

The authors ignored all my comments and did not revise.
Therefore, I can not recommend this article for publication.

Experimental design

The methodology is not rigorous. PRISMA should be followed.

Validity of the findings

The findings are not valid due to lack of rigorous methodology.

Reviewer 3 ·

Basic reporting

no comment

Experimental design

no comment

Validity of the findings

In my opinion, a statistical analysis of the results should be included in the article, to show whether the differences between models are significant.

Additional comments

no comment

·

Basic reporting

Clear and unambiguous English is used in most of the papers. However, there are some minor errors in English in some parts, as follows,
"Secondly, is the Physics-Based (PBs) Algorithms, which aim to mimic physical rules in their search process."

The review is very interesting because it presents an wide range of various methods classified as traditional and meta-heuristic (MH) methods in feature selection.

This review provides a systematic analysis of MH technologies used in feature selection between 2015 and 2022, thus providing an insight into recent developments in the field.

Experimental design

The research methodology in this survey provides a comprehensive exploration of meta-heuristic (MH) methods, and references to them are appropriately cited.
The conclusion of this review is appropriately described. It shows the potential for further research into the MH techniques in text classification in order to refine feature selection in a variety of application domains.

Validity of the findings

The conclusions of this review are appropriately described. It shows the potential for further research into MH techniques in text classification in order to refine feature selection in various application domains.

Additional comments

To further develop this area of research in the future, we would like you to indicate unsolved issues and interesting research challenges in these FS researches based on the MH.

---

## Round 0.3 · accepted · Accept

The current revision addresses the minor concerns raised by the reviewers in the previous round (i.e. statistical testing and fleshing out future directions). These revisions are satisfactory and I'm therefore happy to recommend acceptance of the paper in its current form.